# Catalytic site flexibility facilitates the substrate and catalytic promiscuity of *Vibrio* dual lipase/transferase

Chongyang Wang[1,2,3], Changshui Liu[1,2], Xiaochuan Zhu[1], Quancai Peng[4] & Qingjun Ma [1,2,3,4] ✉

Although enzyme catalysis is typified by high specificity, enzymes can catalyze various substrates (substrate promiscuity) and/or different reaction types (catalytic promiscuity) using a single active site. This interesting phenomenon is widely distributed in enzyme catalysis, with both fundamental and applied importance. To date, the mechanistic understanding of enzyme promiscuity is very limited. Herein, we report the structural mechanism underlying the substrate and catalytic promiscuity of *Vibrio* dual lipase/transferase (VDLT). Crystal structures of the VDLT from *Vibrio alginolyticus* (ValDLT) and its fatty acid complexes were solved, revealing prominent structural flexibility. In particular, the "Ser−His−Asp" catalytic triad machinery of ValDLT contains an intrinsically flexible oxyanion hole. Analysis of ligand-bound structures and mutagenesis showed that the flexible oxyanion hole and other binding residues can undergo distinct conformational changes to facilitate substrate and catalytic promiscuity. Our study reveals a previously unknown flexible form of the famous catalytic triad machinery and proposes a "catalytic site tuning" mechanism to expand the mechanistic paradigm of enzyme promiscuity.

In textbooks, enzymes are referred to as biocatalysts with exquisite specificity. Nevertheless, many enzymes can act on different substrates or even catalyze different reaction types at the same active site. This interesting phenomenon is known as enzyme promiscuity, with the former often called substrate promiscuity or ambiguity, and the latter catalytic promiscuity[1,2]. Indeed, enzyme promiscuity is not rare but widely distributed in living organisms[1]. A survey of the *Escherichia coli* genome showed that 37% of the enzymes exhibit promiscuous activity and are responsible for 65% of in vivo metabolic reactions, implying that promiscuity could be a general characteristic of a large portion of total enzymes[3]. Nowadays, enzyme promiscuity has become a modern term in enzymology and has aroused wide interest for its importance in the evolution of novel enzyme functions in nature, the generation of new biotechnical catalysts, and the increasing multidrug

resistance of pathogens[4–7]. Knowing how promiscuity is achieved at the molecular level is essential for better understanding and utilization of promiscuity[8]. However, compared to the great knowledge of enzyme specificity, the structural and mechanistic understanding of promiscuity is very limited.

*Vibrio* dual lipases/transferases (VDLTs, e.g., VPA0226 of *Vibrio parahaemolyticus*[9–11], lec/VC_A0218 of *Vibrio cholera*[12], and VvPlpA of *Vibrio vulnificus*[13]) are a class of virulence factors secreted by *Vibrio* pathogens to disrupt host membrane structures. For a long time, their biological functions have been attributed to their lipase activity. In vitro characterization showed that VDLTs exhibit phospholipase A2, lysophospholipase and esterase activities, to hydrolyze diverse lipidic substrates[10,13–18]. Interestingly, a recent study by Chimalapati et al. showed that VPA0226, a type 2 secreted lipase that helps

[1]CAS and Shandong Province Key Laboratory of Experimental Marine Biology, Institute of Oceanology, Chinese Academy of Sciences, Qingdao, China. [2]Laboratory for Marine Biology and Biotechnology, Qingdao National Laboratory for Marine Science and Technology, Qingdao, China. [3]University of Chinese Academy of Sciences, Beijing, China. [4]Center for Ocean Mega-Science, Institute of Oceanology, Chinese Academy of Sciences, Qingdao, China. ✉e-mail: qma@qdio.ac.cn

*V. parahaemolyticus* escape from the host cell, not only exhibited the traditional lipase activity but also could transfer the acyl chain from various host lipids to cholesterol[11]. This glycerophospholipid: cholesterol acyltransferase activity (EC2.3.1.-), differing in the first digit of the EC numbering from its canonical lipase activity (EC3.1.1.-), occurs at the same active site. These evidences collectively indicate that VDLTs are able to act on various substrates and catalyze different types of reactions. Therefore, VDLTs are enzymes with unusual dual substrate and catalytic promiscuity, which could benefit pathogens in handling diverse host lipid molecules in a proper manner. How these lipases exhibit this interesting dual promiscuity with a single active site is intriguing.

VDLTs contain an N-terminal domain (NTD) of unknown function and a C-terminal catalytic domain belonging to the SGNH/GDSL hydrolase family, which is named after the four characteristic catalytic residues (Ser, Gly, Asn, and His)[19]. Previously, we solved the *apo* structure of VvPlpA from *V. vulnificus*[20]. This structure reveals an overall architecture with the two domains closely packed and a unique Ser−His−chloride catalytic triad[20,21], although other members of VDLTs are expected to have a more common Ser−His−Asp/Glu triad (Supplementary Fig. 1a). Noticeably, in the *apo* structure of VvPlpA, the catalytic apparatus is in an inactive state due to an improperly formed oxyanion hole. Moreover, the proposed substrate-binding pocket is also shielded from the surrounding solvent. Obviously, conformational rearrangement would occur during the catalysis process to enable substrate entry and catalytic site activation. We expect that ligand-bound structures of VDLTs would better reveal catalysis-relevant conformational states, thus aiding the mechanistic understanding of their dual promiscuity.

Herein, we report crystal structures of the VDLT from *V. alginolyticus* (ValDLT), alone and in complex with a series of fatty acids, along with mutagenesis and enzymatic data. We discovered prominent flexibility in the ValDLT structure, even in the catalytic site, which facilitates dual promiscuity via a "catalytic site tuning" mechanism. In general, the results broaden our mechanistic understanding of enzyme promiscuity.

## Results

### ValDLT harbors a catalytic site with intrinsically flexible oxyanion hole

The *apo* ValDLT was crystallized in the P2$_1$ space group with two protein monomers in one asymmetric unit. ValDLT shares a similar overall structure with VvPlpA, with the function-unknown NTD closely packed on the catalytic SGNH domain (Fig. 1a). However, unlike the occluded active site in VvPlpA, the active site in ValDLT is readily accessible from the solvent region. A polyethylene glycol (PEG) molecule derived from the crystallization buffer could be modeled in the substrate-binding pocket of each monomer.

Superposition of the structures of VvPlpA and ValDLT monomers A and B showed that the majority of these structures overlapped well, with a pairwise RMSD of 0.3−0.6 Å for the aligned Cα atoms (Fig. 1b). Nevertheless, the superposition also identified conformational heterogeneity in both terminal regions and some additional regions, such as loop$^{\beta3-\beta4}$ ($^{81}$WWSSVSFKNM$^{90}$) and loop$^{\beta8-\alpha5}$ ($^{202}$VGGAAGENQYIALT$^{215}$), indicating flexibility in these structural segments (Fig. 1b). Notably, loop$^{\beta8-\alpha5}$ is located at the opening of the substrate-binding pocket. It exhibits a defined conformation in monomer B; however, in monomer A, most of it (residues 204 to 214) is completely disordered, lacking defined electron density to trace. Checking the crystal packing, we found that this loop in monomer A is exposed in the bulky solvent, thus allowing its free motion. We also noticed that this loop contains a significant proportion of small amino acid residues, indicating its flexible nature.

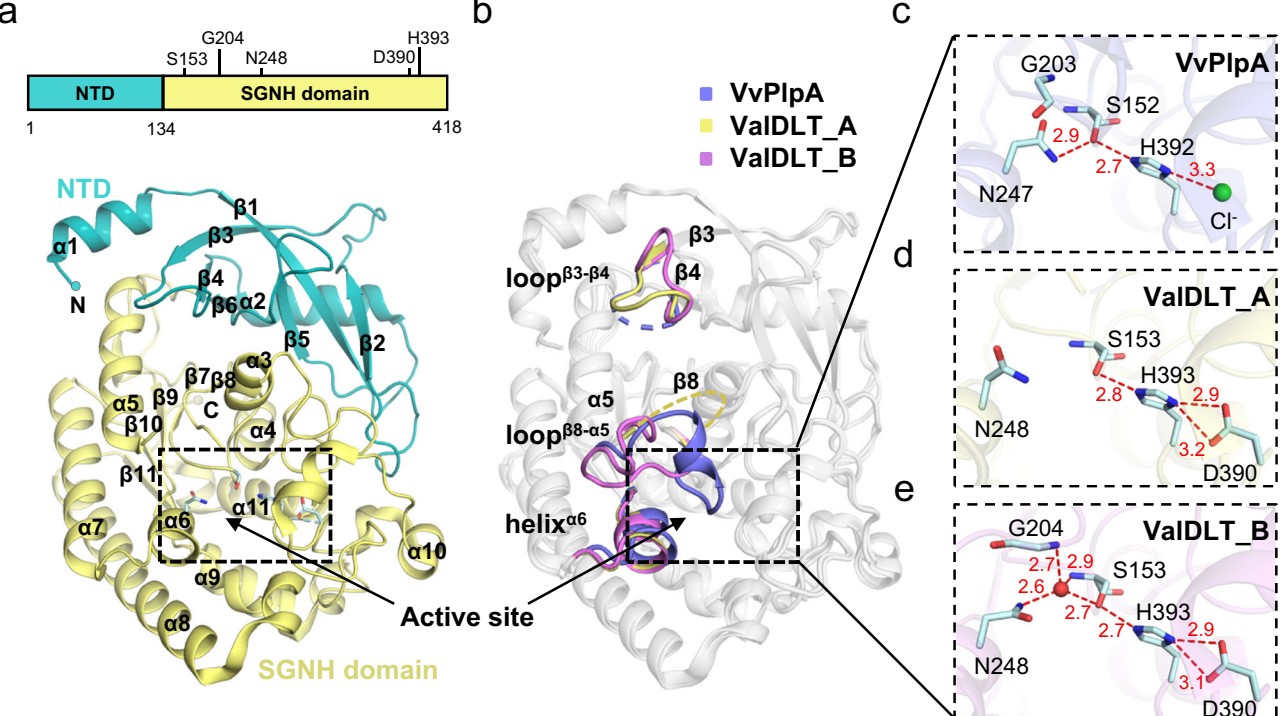

**Fig. 1 | Overall structure and catalytic site of *apo* ValDLT. a** Overall structure of *apo* ValDLT (representing monomer A). The function-unknown NTD, the C-terminal SGNH domain and the secondary structural elements are indicated. The catalytic residues are shown as sticks, with their positions indicated (upper panel). **b** Superposition of the overall structures of ValDLT monomer A (ValDLT_A), monomer B (ValDLT_B) and VvPlpA. The regions with significant conformational differences are indicated. **c−e** Catalytic sites of VvPlpA, ValDLT_A and ValDLT_B, respectively. Residues are shown as sticks, and water molecules are shown as red spheres. Hydrogen bonds are represented as red dashed lines, with the distance (Å) indicated. The oxyanion hole residue G204 is disordered in ValDLT_A and is not shown in **d**.

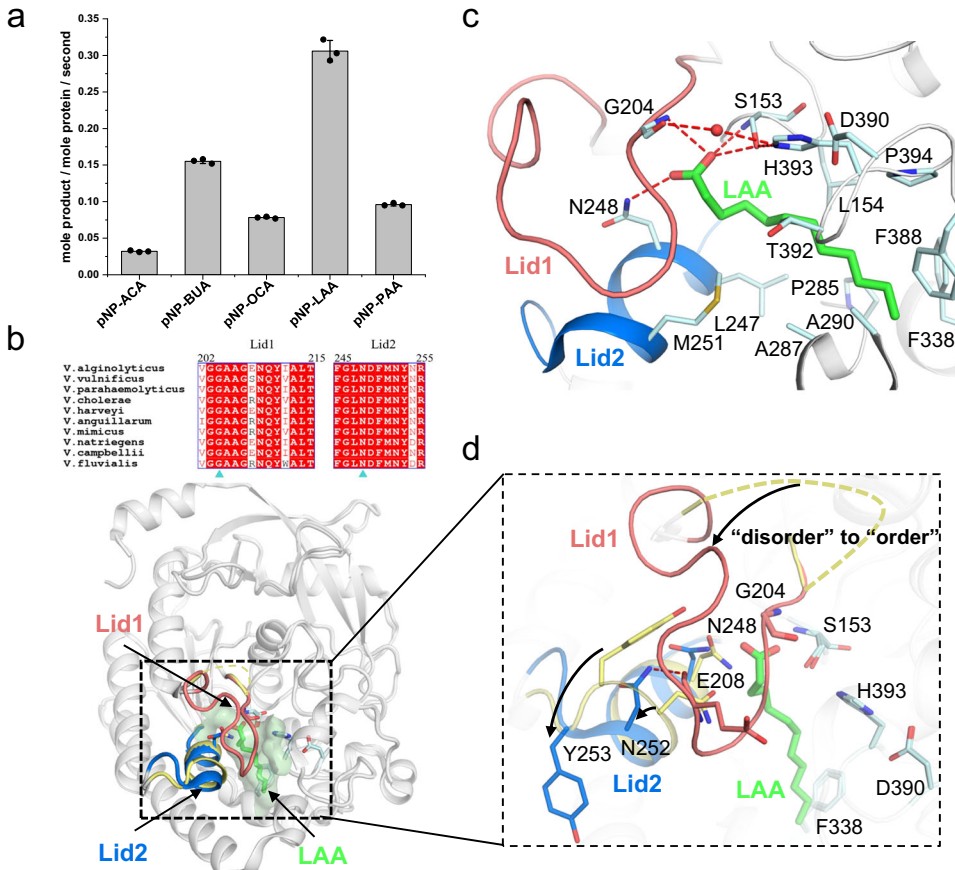

**Fig. 2 | Esterase activity of ValDLT and its structure in complex with LAA.**
**a** Specific esterase activity on p-nitrophenyl derivatives of fatty acids. pNP-ACA: p-nitrophenyl acetic acid, pNP-BUA: p-nitrophenyl butanoic acid, pNP-OCA: p-nitrophenyl octanoic acid, pNP-LAA: p-nitrophenyl lauric acid, pNP-PAA: p-nitrophenyl palmitic acid. Data are the mean ± SD of three independent measurements. **b** Superposition of the overall structures of the ValDLT/LAA complex and *apo* ValDLT (monomer A). LAA and the two lid regions in ValDLT are indicated.

Sequence alignment of the two lids is shown, with the oxyanion hole residues G204 and N248 indicated by blue triangles. **c** The active site of the ValDLT/LAA complex. **d** Superposition of the active sites of the ValDLT/LAA complex and *apo* ValDLT. Conformational changes are indicated. The residues and ligands are shown in stick representation. The water molecule is shown as a red sphere. Hydrogen bonds are shown as red dashed lines. Source data are provided with this paper.

Remarkably, superposition also clearly revealed diverse configurations in the catalytic sites of these structures. As mentioned, the catalytic site of *apo* VvPlpA contains a unique Ser−His−chloride triad, however, the oxyanion hole residues are in contact and thus in an inactive state (Fig. 1c). In comparison, ValDLT contains a canonical Ser−His−Asp catalytic triad composed of residues S153, H393 and D390, with the oxyanion hole contributed by the main-chain NH groups of S153 and G204 and the side-chain amide group of N248. In both monomers A and B, the catalytic triad is well structured to form a functional charge relay system as seen for VvPlpA (Fig. 1d, e), but the oxyanion hole configurations differ from each other, and from that of VvPlpA. In monomer A, the oxyanion hole is ambiguously defined, as G204 is located at the disordered region of loop$^{β8-α5}$ (Fig. 1d). In monomer B, the same region is in an ordered conformation that brings G204 into a proper oxyanion-hole forming position, due to a $Mg^{2+}$ cation that bridges E208 and D263 (Supplementary Fig. 2a). Furthermore, another oxyanion hole residue, N248 in monomer B, differs in the side-chain conformation from that of monomer A. Unlike those in VvPlpA and monomer A, the oxyanion hole in monomer B appears to be well-formed, with a water molecule occupying the putative oxyanion hole position, stabilized by the hydrogen bonds from all three oxyanion hole residues, S153, G204 and N248 (Fig. 1e). Based on the comparison, we conclude that the catalytic site of ValDLT is composed of a relatively fixed catalytic triad but intrinsically flexible oxyanion hole residues. Interestingly, it has been proposed that SGNH family

proteins can exhibit active site flexibility[19], with some reports including flexibility of the catalytic residues. For example, the acetyl-xylan esterase from an Arctic marine bacterium has the catalytic His and Asp in a long flexible active site loop, which is expected to play a role in cold adaptation of this enzyme[22]. In the bacterial acetylcholinesterase structure, the catalytic Ser can adopt productive and nonproductive conformations depending on substrate binding[23]. Nevertheless, the large-scale flexibility of oxyanion hole residues, as seen in ValDLT, has rarely been reported[24].

## The active site undergoes dramatic conformational changes for catalysis

To understand how the flexible active site is constituted in catalysis, we sought to solve the ligand-bound structures of ValDLT. ValDLT hydrolyzed an array of p-nitrophenyl derivatives of fatty acids (Fig. 2a). We solved the crystal structure of ValDLT in complex with lauric acid (LAA), whose p-nitrophenyl derivative showed the highest specific activity. This enzyme-product complex structure is supposed to reveal the conformational information related to the lipase/esterase activity of ValDLT.

The ValDLT/LAA complex was crystallized in the same space group as that of the *apo* form, with a similar space arrangement of the two protein molecules *per* asymmetric unit. The comparison of monomers A and B with those in the *apo* form is shown in Supplementary Fig. 3. In the active site region, ValDLT/LAA monomer A shows

clear structural differences in reference to *apo* monomer A. In contrast, the active site region of ValDLT/LAA monomer B is structurally similar to that of *apo* monomer B, with loop$^{\beta 8-\alpha 5}$ in an open conformation and the catalytic site residues well superposed. The LAA molecule was only identified in the substrate-binding pocket (hereafter designated acyl-binding pocket) of monomer A (Fig. 2b). As no LAA was identified in monomer B, hereafter, our structural analysis was based on monomer A only, unless otherwise specified.

The hydrocarbon tail of LAA displays an elongated conformation, interacting extensively with the hydrophobic wall of the acyl-binding pocket. The hydrocarbon chain of LAA appears to occupy only part of the space of the acyl-binding pocket, with its tail flanked by F338 and F388 (Fig. 2c). The carboxyl group of LAA is located at the opening of the pocket, forming hydrogen bonds to the NH groups of S153 and G204 with one of its oxygen atoms, and to the amide group of N248 with the other (Fig. 2c). In this way, the oxyanion hole adopts a functional configuration. In addition, a water molecule is coordinated by the catalytic residues H393 and S153 and could act as the nucleophile attacking the enzyme-acyl intermediate in the catalysis process.

Compared to the *apo* structure (Fig. 2d), the most prominent difference occurs in loop$^{\beta 8-\alpha 5}$, which undergoes a disorder-to-order conformational change to cover the acyl-binding pocket, with G204 interacting with the carboxyl group of LAA. Accompanying this ordering event, a segment containing helix$^{\alpha 6}$ and the flanking short coils (including residues $^{245}$FGLNDFMNYNR$^{255}$), which is located at the opening and wall of the acyl-binding pocket, also reforms its conformation in a twisting-like manner; otherwise, this segment would clash with the newly ordered loop in space. In this new conformation, Y253 now flips to the opposite side, while N252 shifts its position to form a hydrogen bond with the main-chain oxygen of E208 of loop$^{\beta 8-\alpha 5}$, as well as a hydrogen bond with the oxyanion hole residue N248, which itself also rotates to form a hydrogen bond with the LAA carboxyl oxygen. Loop$^{\beta 8-\alpha 5}$ and the aforementioned segment contain the oxyanion hole residues and undergo ligand-induced conformational changes to cap on the acyl-binding pocket, so they are conveniently named Lid1 and Lid2, respectively. From the structural point of view, these two lids are supposed to play important roles in catalysis, which is further supported by their sequence conservation among VDLTs (Supplementary Fig. 1).

We sought to test the role of lid conformational changes in catalysis. Mg$^{2+}$ can bridge residue E208 of Lid1 to D263 of the main protein body and is thus capable of modulating the conformation of Lid1 (Supplementary Fig. 2a). We hypothesized that metal ions would affect enzymatic activity if the Lid1 conformation matters. Indeed, various divalent metal ions could enhance the enzymatic activity of ValDLT, among which Mn$^{2+}$ showed the largest effect (Supplementary Fig. 2b). In the presence of Mn$^{2+}$, the values of $k_{cat}$ and $K_M$ for ValDLT esterase activity were increased 8.8- and 1.7-fold, respectively, resulting in a higher $k_{cat}/K_M$ (5.2-fold) (Supplementary Fig. 2c). The mutants of metal-chelating residues (ValDLT$^{E208S}$ and ValDLT$^{D263A}$) lost their response to Mn$^{2+}$, confirming that the metal effect on activity originated from Lid1 conformation modulation. The activity enhancement even let us consider the physiological role of metals in ValDLT function. However, the binding affinity of metals to ValDLT is only at the mM level, as estimated from a titration enzymatic assay (Supplementary Fig. 2d), so they are unlikely to act as physiological allosteric agents. Nevertheless, using the metal effect as a probe, we show that the conformational changes at the lid region play an important role in ValDLT catalysis.

## Binding of long-chain fatty acids reveals an alternative conformation for hybrid transferase/hydrolase activity

In a recent study, VPA0226 was reported to have glycerophospholipid: cholesterol acyltransferase activity in addition to classical phospholipase activity[11]. A schematic description of this acyl-transferring reaction is shown in Supplementary Fig. 4. We performed a similar liposome assay and showed that VDLTs, including ValDLT, VvPlpA, VPA0226 and lec/VC_A0218, could also transfer the oleoyl group of cardiolipin (18:1) to cholesterol (Fig. 3a). This result is not surprising, given the high-sequence identity among VDLTs (≥60%) (Supplementary Fig. 1). Mutating the catalytic base His393 to Ala (ValDLT$^{H393A}$) abolished the transferase activity (Fig. 3a), indicating that the transferase activity shares its active site with the traditional phospholipase activity.

To understand the transferase mechanism of ValDLT, we attempted to crystallize it with cardiolipin (18:1) and/or cholesterol but failed. As an alternative, we solved the ValDLT structure in complex with oleic acid (OLA), which is the acyl group of cardiolipin (18:1). This structure is believed to partially reveal the conformational information of the acyl-enzyme intermediate related to the transferase activity. The ValDLT/OLA crystal shares a similar crystal packing with those of the *apo* ValDLT and ValDLT/LAA, containing two monomers in one asymmetric unit. The structural comparison of monomers A and B with that of the *apo* and LAA bound forms is shown in Supplementary Fig. 3. Similar to LAA, OLA is accommodated only in the acyl-binding pocket of monomer A (Fig. 3b). If not specified, the following structural analysis will refer to monomer A only.

The long hydrocarbon chain of OLA adopts a bent conformation, passing by F338 and approaching the bottom residues W185 and M282 of the pocket. The carboxyl group interacts extensively with the residues located at the mouth of the pocket: one oxygen forms hydrogen bonds with the main-chain NH group and side-chain hydroxyl group of S153, while the other can form hydrogen bonds to the main-chain NH groups of G204 and A205, as well as the side-chain NH2 group of N252. However, the carboxyl group has no interaction with the oxyanion residue N248, whose side chain is oriented away from the oxyanion hole forming position. Noticeably, Lid1 seems to enclose the acyl-binding pocket, occluding it from the bulky solvent. However, a water molecule is captured in the active site cavity, coordinated by E208 of Lid1 and the catalytic residues S153 and H393 via hydrogen bonds, and this water molecule may also form a hydrogen bond with the OLA carboxyl group.

In reference to *apo* ValDLT (Fig. 3c and Supplementary Fig. 3), Lid1 of ValDLT/OLA undergoes a disorder-to-order conformational transition to cap the acyl-binding pocket, while Lid2 barely changes its conformation. Instead, helix$^{\alpha 3}$ ($^{159}$NIFNASQWRFP$^{169}$, in coil conformation in the complex) and its neighboring residues exhibit a clear positional shift, which could be attributed to repelling by the ordered Lid1. In addition, F338, acting as an inner lid of the acyl-binding pocket, rotates aside to allow the long hydrocarbon chain of OLA to enter deeper. Interestingly, similar tunnel adaptation of the Phe residue has also been observed in the SGNH family enzyme mammalian lipopolysaccharide detoxifier[25]. We also compared the LAA- and OLA-bound structures (Fig. 3d and Supplementary Figs. 3 and 5), which showed that the hydrocarbon chains of the two fatty acids exhibit different conformations and occupy different areas of the acyl-binding pocket. Moreover, their carboxyl heads differ in position and orientation. Most interestingly, Lid1 of ValDLT undergoes very different disorder-to-order conformational changes in the two complexes. Furthermore, the two complexes also differ in the conformations of Lid2, helix$^{\alpha 3}$ and the inner Phe lid. We were surprised to find that OLA remodels the active site in a distinct manner from LAA, as the two fatty acids are chemically analogous. We also solved the structures of ValDLT in complex with arachidonic acid (ARA) and docosahexaenoic acid (DHA), two other long-chain fatty acids (LCFAs) that can be transferred to cholesterol[11]. ARA and DHA are buried slightly deeper in the acyl-binding pocket than OLA due to their longer hydrocarbon chains, and in general, the two complexes share similarity with ValDLT/OLA in active site conformation and ligand interactions (Supplementary Figs. 3, 5, and 6). All three LCFA-bound structures suggest that ValDLT catalyzes substrates

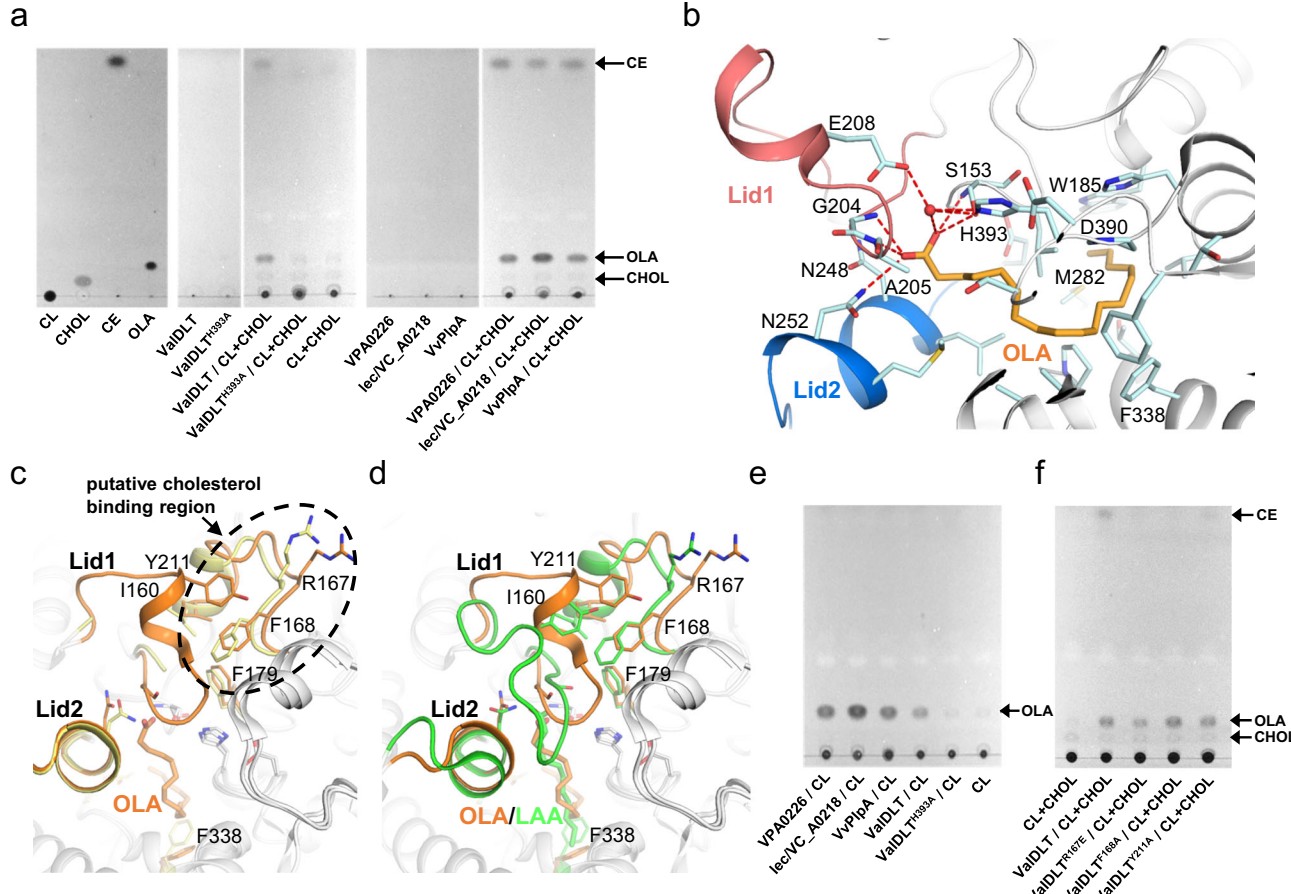

**Fig. 3 | The transferase/lipase activity of ValDLT and the structure of the ValDLT/OLA complex. a** Thin-layer chromatography of the lipids extracted from the glycerophospholipid: cholesterol acyltransferase assay. CL: cardiolipin (18:1); CHOL: cholesterol; CE: cholestreyloleate; OLA: oleic acid. **b** The active site of the ValDLT/OLA complex. The residues and OLA are shown as sticks. Water is shown as a red sphere. Hydrogen bonds are represented as red dashed lines. **c** Superposition of the active sites of the ValDLT/OLA complex and *apo* ValDLT. **d** Superposition of the active sites of the ValDLT/OLA complex and ValDLT/LAA complex. **e** The same assay as in **a**, but omitting cholesterol, indicated the hydrolase activity of VDLTs. **f** The same assay as in **a** for the mutants of the putative cholesterol binding site. Source data are provided with this paper.

with long acyl chains via a different conformational pathway from those with short or medium acyl chains.

We attempted to understand the transferase activity based on these LCFA complex structures. At first glance, it was confusing to find a water molecule close to the catalytic residues, which implies a potential hydrolysis reaction of the acyl-enzyme intermediate. Further inspection of the liposome assay identified a band assumed to be oleic acid, which could be the hydrolysis product of cardiolipin (18:1) (Fig. 3a). Afterward, we performed a similar assay but omitted cholesterol, and the hydrolysis product oleic acid was clearly observed (Fig. 3e). It is likely that the hydrolase and transferase activities on cardiolipin (18:1) co-occurred through the same conformational state as shown in the LCFA complex structures. Such a hydrolase/transferase co-occurrence sounds unusual but is not unprecedented. Several enzymes were reported to concurrently possess hydrolase and transferase activities, and are called promiscuous hydrolase/transferase enzymes[26]. With respect to ValDLT, cholesterol and water may compete to accept the acyl group from the same acyl-enzyme intermediate, leading to transferase and hydrolase activities, respectively.

We also investigated the cholesterol binding site. It is not obviously indicated in the LCFA complex structures, where the active site is in an occluded state. Previous studies on the cholesterol binding site showed that a consensus binding motif of cholesterol is hard to define, but aromatic residues such as Tyr, Phe, and Trp, as well as positively charged residues such as Arg, Lys, and His, are often

involved in cholesterol binding[27]. Guided by these rules, we scrutinized the active site and speculated that a cluster of residues close in space, including F179 close to the catalytic residues, I160, R167 and F168 at the helix$^{\alpha 3}$ region, and Y211 in Lid1, could contribute to a potential cholesterol binding site (Fig. 3c). These residues are conserved in VDLTs (Supplementary Fig. 1), supporting a functional role. Remarkably, these residues undergo significant conformational changes upon LCFA binding, indicating their conformational flexibility, which would give the potential of further structural remodeling for cholesterol binding. We prepared mutants of this site, including ValDLT$^{R167E}$, ValDLT$^{F168A}$ and ValDLT$^{Y211A}$, which were well expressed and folded (Supplementary Fig. 7). For these mutants, the transferase activity was abolished or dramatically decreased, while the hydrolase activity was retained (Fig. 3f), suggesting that these mutations affected the transferase activity by disrupting cholesterol binding, thus supporting our proposal about the cholesterol binding site. Nevertheless, a cholesterol-enzyme complex structure would be desired to reveal the exact cholesterol binding mode of VDLTs.

## The flexible oxyanion hole might be an adaptive feature for promiscuous catalysis

To obtain a comprehensive view of the flexibility and conformational changes of ValDLT, we performed a systematic comparative study using all available ValDLT structures, including the *apo* form and its complexes with LAA, OLA, ARA and DHA. Luckily, the similar molecular

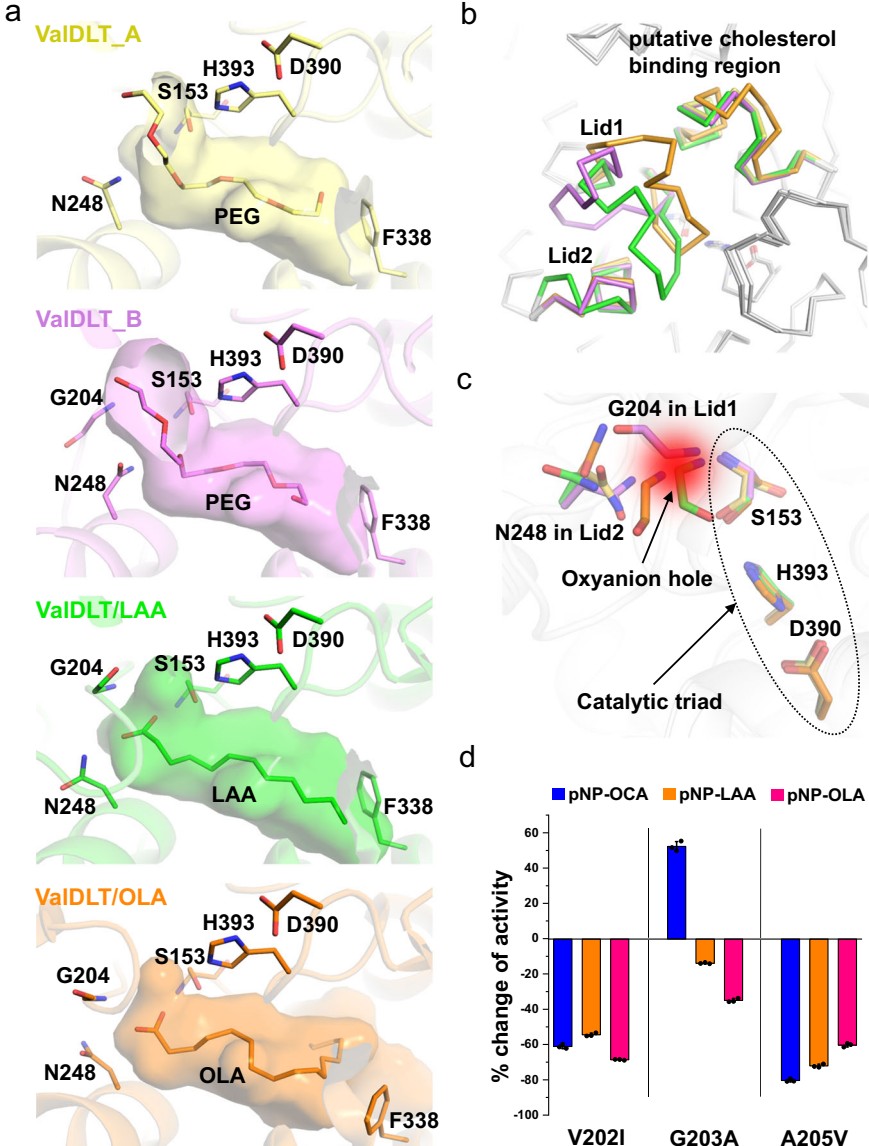

**Fig. 4 | The catalytic site flexibility of ValDLT and its relatedness to promiscuous catalysis. a** Four representative binding modes of ligands in the acyl-binding pocket of ValDLT. The catalytic site residues, Phe lid and ligands are shown as sticks. **b** Superposition of overall ValDLT structures in **a**. Regions with conformational differences are colored with the same scheme in **a**. **c** Superposition of the catalytic sites of all ValDLT structures in **a**. The residues are colored as in **a**. **d** Mutation of the oxyanion hole region resulted in substrate-dependent effects on esterase activity. Data are the mean ± SD of three independent measurements. Source data are provided with this paper.

arrangement of all crystals enabled us to better distinguish the "real" structural variations from the crystal packing effects. We focused on the active site region, identifying four representative conformational scenarios (Fig. 4a–c). The acyl-binding pocket of ValDLT is a broad and hydrophobic space to accommodate the acyl chain that is subject to cleavage or transfer (Fig. 4a). Remarkably, the catalytic nucleophile (S153), where the chemistry occurs, is located at the opening of the pocket. The fatty acid ligands are accommodated in this acyl-binding pocket in different poses, displaying different interactions with the surrounding residues (Fig. 4a and Supplementary Fig. 5). The hydrocarbon tails of OLA, ARA, and DHA, which are unsaturated long-chain fatty acids, all adopt a bent conformation to occupy a large portion of the pocket. Their tails follow a similar tracking path with certain differences in zigzag detail, and their carboxyl groups are in the same position with similar orientation. In contrast, LAA, a saturated medium-chain fatty acid, has its hydrocarbon tail in an extended conformation, in a space different from the previously discussed fatty acids; the

carboxyl group of LAA is located at a position obviously different from that of the former three fatty acids. Accordingly, the active site region of ValDLT, including the two lids and some other residues, which have substantial conformational flexibility, can be dramatically remodeled in distinct ways to interact with these ligands. Uniquely, the catalytic site of ValDLT contains intrinsically flexible oxyanion hole residues, which could vary in conformation at the side-chain level (N248) or the main-chain level (G204) to sample a larger space in constituting the oxyanion hole than fixed conformations, thereby better stabilizing the transition states that might vary in a substrate-dependent manner (Fig. 4c).

To shed more light on the role of oxyanion hole flexibility, we designed a microperturbation experiment, in which the residues at the oxyanion hole region were mutated to residues with slightly bulkier side chains to limit the local flexibility with minimal disturbance of the local structure. We prepared the mutants ValDLT[V202I], ValDLT[G203A], ValDLT[G204A], and ValDLT[A205V] and measured their activities on pNP-

OCA, pNP-LAA and pNP-OLA (Fig. 4d). These substrates are analogous compounds with the same nitrophenyl group and differ only in their acyl chains. During hydrolysis, their acyl chains would be buried in the acyl-binding pocket of ValDLT. If the acyl-binding modes position their oxyanion transition states at the same site at the mouth of the acyl-binding pocket, then the mutations at the oxyanion hole region (here on Lid1) would have comparable effects on enzyme activity. ValDLT$^{G204A}$ abolished activity on all substrates, indicating the essentiality of G204 as the oxyanion hole residue. Other mutations altered the enzyme activity in a substrate-dependent manner (Fig. 4d). For example, compared to that of the WT, the activities of ValDLT$^{G203A}$ on pNP-LAA and pNP-OLA were decreased, while its activity on pNP-OCA was increased by 52%. Both ValDLT$^{V202I}$ and ValDLT$^{A205V}$ exhibited decreased activities on pNP-LAA and pNP-OLA, but to different extents (Fig. 4d, blue and pink bars). The substrate-dependent effects of these mutations may reflect the different binding modes of the acyl chains and the resultant different positioning of the oxyanion, which would require differential oxyanion hole configurations for optimal catalysis, somewhat explaining the flexibility of the oxyanion hole in ValDLT. Inspired by the increased activity of ValDLT$^{G203A}$ on pNP-OCA, we even speculate that a preorganized oxyanion hole configuration could have evolved in VDLTs to better catalyze a given substrate, albeit with a trade-off of lower activities on other substrates. However, because of its flexible oxyanion hole, VDLT can readily satisfy the conformational demands of a diverse range of substrates, maintaining a balance between substrate diversity and catalytic efficiency. From this perspective, the flexible oxyanion hole seems to be an adaptive feature of VDLT to facilitate promiscuous catalysis.

## Discussion

VDLTs are important virulence factors of *Vibrio* pathogens, exhibiting unusual dual substrate and catalytic promiscuity. Here, we explored the underlying mechanism by structural and enzymatic approaches. We discovered some uncommon and interesting features related to the promiscuous catalysis of VDLTs, offering mechanistic insight into enzyme promiscuity.

Flexibility is a notable feature of VDLT structures. Previously, VvPlpA was reported to have a clashed oxyanion hole in the ligand-free state, implicating catalytic site reconstitution during catalysis, and furthermore, additional flexibility in the other region of the active site was also identified when analyzing one mutant structure[20]. Here, comparing structures of ValDLT in *apo* and ligand-bound states clearly demonstrates the flexibility in the active site, which undergoes dramatic conformational changes upon fatty acid binding, allowing the oxyanion hole to be reconstituted to a catalytically competent state. The metal effects on the enzymatic activity of ValDLT support the important role of conformational changes in catalysis. Notably, although we observed two distinct conformations in the fatty acid-bound structures, these conformations do not seem to be directly suitable for binding other ligands, such as glycerophospholipid substrates or cholesterol. This indicates that other conformational changes remain to be discovered. Combining all structural evidences, it is clear that flexibility is an important feature of VDLTs, enabling them to perform dynamic conformational changes in the catalytic process. Owing to the functional importance of VDLTs in *Vibrio* infection, they have been considered potential antivirulence drug targets. Their structural flexibility needs to be fully considered in the practice of structure-based drug design.

An even more prominent and unique feature of ValDLT is that its catalytic site contains an intrinsically flexible oxyanion hole. The catalytic site of VDLT belongs to the very famous "catalytic triad" machinery, whose structure and working mechanism have been well-studied. It is characterized by a "nucleophile−base−acid" triad, usually Ser−His−Asp or variant forms, accompanied by an oxyanion hole contributed by amide groups[28]. Since its initial discovery in chymotrypsin and subtilisin fifty years ago[29,30], it has been widely found in numerous hydrolases and transferases, which perform covalent catalysis. Based on structural studies, the binding sites of these enzymes can often undergo conformational changes upon ligand binding. In contrast, their catalytic sites are usually maintained in a stable conformation in most cases, although they can occasionally display limited side-chain flexibility[24]. The rigidity of the catalytic site may reflect the requirement for precise stabilization of the transition state[31]. Interestingly, the oxyanion hole residues (G204 and N248) of ValDLT are located in flexible lids, exhibiting prominent conformational freedom not only at the side-chain level but also at the main-chain level (Fig. 4c). To our knowledge, such high flexibility of oxyanion residues has not been reported before. To clarify, oxyanion hole residues can undergo significant conformational changes during enzyme maturation or activation processes as a means to control enzyme activity[32–34], but flexibility of the oxyanion hole residues is no longer needed in mature/activated enzymes. Given the essential role of the oxyanion hole in stabilizing the transition state, its flexibility is expected to have a profound effect on catalysis. Therefore, we define the catalytic triad with an inherently flexible oxyanion hole as a special variant of the "catalytic triad" class. The discovery of this flexible variant suggests that the "ancient" and well-studied catalytic triad machinery may have more dynamic features than we previously thought, which should be considered in future studies.

In this study, we experimentally confirmed the dual substrate and catalytic promiscuity in VDLTs. Based on structural analysis, we propose a connection between the dual promiscuity and structural flexibility, highlighting the roles of flexible oxyanion hole residues. Most importantly, the catalytic nucleophile (Ser) of VDLT is located at the mouth of its acyl-binding pocket. For catalysis, the substrate acyl chain that undergoes cleavage/transfer has to be completely buried in the confined space of this pocket for its −COO- moiety to approach the nucleophilic serine. We propose that acyl chains with different lengths and saturations may bind differently within the pocket, and thus, the −COO- ester groups may vary in position and orientation in a substrate-dependent manner; similar conformational variations would also be expected for the oxyanion that forms in this -COO- ester group upon nucleophilic attack. This proposal is well supported by the comparative analysis of ValDLT structures and the oxyanion hole micro-perturbation experiment (Fig. 4). Compared to the canonical preorganized oxyanion hole residues, flexible oxyanion hole residues have an advantage in fine-tuning their conformations to better stabilize oxyanions with various positions and orientations. Moreover, the flexibility in the binding site, which might not be fully revealed in this study, is expected to help bind different substrates. Because of these flexibilities, VDLT is capable of catalyzing a diverse range of substrates. In addition, the structural flexibility contributes to the catalytic promiscuity of VDLT. The catalysis of VDLT, regardless of the lipase or transferase activity, is thought to be composed of two-step reactions: the first step is to transfer the acyl from the substrate to the enzyme, forming a covalent acyl-enzyme intermediate, and the second step is to transfer the acyl from the intermediate to the acceptor molecules (Fig. 5a). The lipase and transferase activities diverge only at the second step, with water and other molecules as the acyl acceptors, respectively. One interesting fact to mention is that the active site region can be distinctly remodeled depending on the binding of different fatty acids (Figs. 3d and 4a–c), which implies that the real acyl-enzyme intermediate may have substrate-dependent conformations. As a major interacting partner of the −COO- ester group, the flexible oxyanion hole residues appear to sense the binding mode of the acyl group by tuning their conformations and dictate the subsequent remodeling of the binding site. Presumably, the resultant conformation of the intermediate would determine the binding preference of cholesterol and thus its competitiveness against water for accepting the acyl group, leading to various balances of transferase and lipase

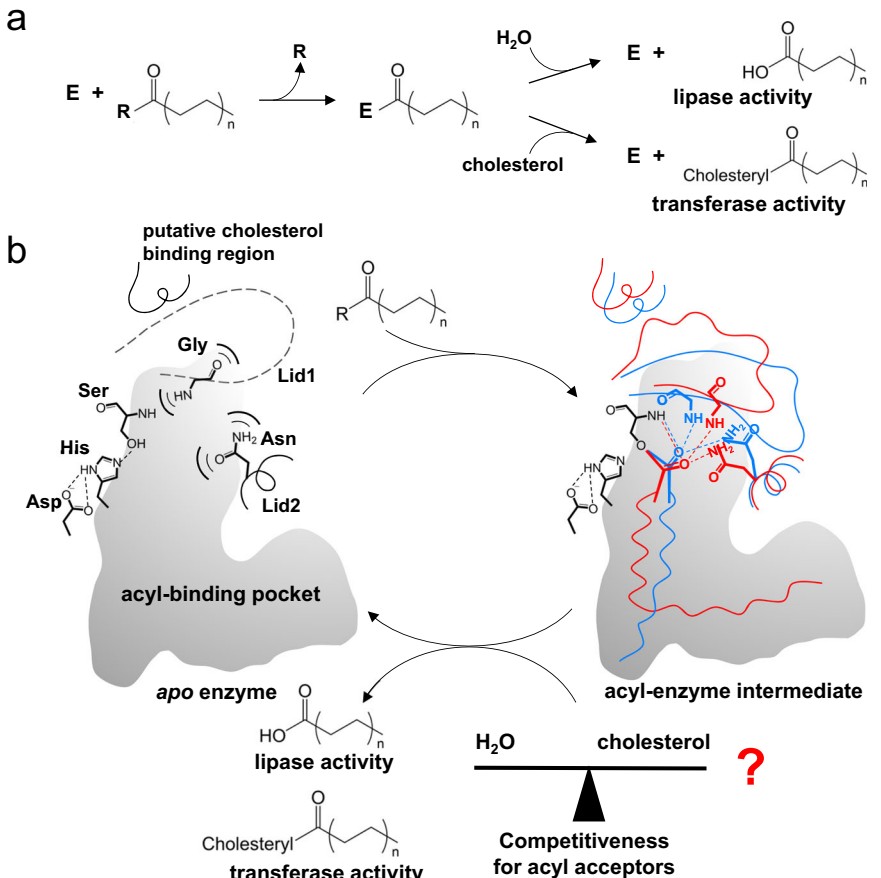

**Fig. 5 | The proposed "catalytic site tuning" mechanism underlying the substrate and catalytic promiscuity of VDLT. a** The catalysis of VDLT (lipase or transferase activity) on lipids takes place in a two-step reaction through an enzyme-acyl intermediate. **b** Structural flexibility facilitates the dual substrate and catalytic promiscuity of VDLT, where catalytic site tuning is at the heart of this proposal. VDLT harbors prominent flexibility in both the binding site and the oxyanion hole of the catalytic site, which helps VDLT bind diverse lipids and stabilize their various activities. In this scenario, the key role of the cleavable/transferable acyl chain should never be underestimated, as it is the source of the initial binding mode difference. There might be a relationship between the acyl type and the lipase/transferase balance, which remains to be clarified. In summary, the flexibility in the active site, particularly that in the oxyanion hole residues, would facilitate the dual substrate and catalytic promiscuity of VDLT. A rough description of the mechanism underlying the dual promiscuity is proposed (Fig. 5). As conformational tuning of the oxyanion hole residues lies at the heart of our proposal, we call this mechanism the "catalytic site tuning" mechanism. We believe this example showcases an intriguing flexibility-based structure-function relationship for enzymes.

Compared to the great knowledge on enzyme specificity, our mechanistic understanding of enzyme promiscuity is very limited[2]. Flexibility has been presumed to play a significant role in promiscuity[35,36]. For example, cytochrome P450 enzymes, famous for their substrate promiscuity, employ flexible active sites to accommodate diverse substrates[37,38]. Tim-barrel fold proteins, such as aromatic prenyltransferases[39] and *Mycobacterium tuberculosis* PriA[40], have active sites located in flexible loop regions, which undergo conformational changes to capture substrates with various shapes. In all these enzymes, the active sites need to be remodeled to enable promiscuous catalysis. Most often, remodeling occurs only at the binding site. Occasionally, the catalytic site itself also needs a reconfiguration for promiscuous catalysis. To our knowledge, the archaeal fructose-1,6-bisphosphate aldolase/phospholipase (FBPA/P), an ancestor Tim-barrel fold protein, is the only well-studied example in this class thus far[41,42]. In FBPA/P, two sets of catalytic machineries are alternatively assembled at the same site to catalyze two successive reactions in the metabolic pathway. Although "active site remodeling" was originally used to describe this mechanism, specifically, we propose the term "catalytic site switching", analogous to a Swiss knife, which reflects the multiple functions and economy of an ancient enzyme. In contrast to FBPA/P, VDLT employs a "catalytic site tuning" mechanism that involves only a single set of catalytic residues. In this mechanism, the oxyanion hole conformation is elegantly adjusted to endow VDLT with dual substrate and catalytic promiscuity, which can be compared to the function of an adjustable wrench. A recent survey of enzyme structures showed that the catalytic sites of a large portion of enzymes display a certain degree of flexibility[24], and the relationship of flexibility to promiscuity is worth attention in future studies. As a secreted virulence factor, VDLT seems to derive two advantages from this mechanism. First, this mechanism enables pathogens to disrupt a broad range of host membranes with diverse lipid components. Second, it provides two different means to modify lipids by controlled lipase/transferase activity, and some of the lipid products may act as signaling molecules to regulate the inflammatory and immune response of the hosts to aid pathogen infection[43]. Therefore, VDLT serves as an excellent example of the structural adaptation of an enzyme to biological functions. In summary, our study reveals flexible

transition states. The acyl-enzyme intermediates may adopt acyl-dependent conformations at the active site (represented by red and blue colors, respectively), which regulates the cholesterol and water competitiveness for accepting the acyl group, resulting in a balance of the transferase and lipase activity. The catalytic triad (Ser–His–Asp) are shown, with the main-chain NH groups of Ser and Gly (on the flexible Lid1) and the side-chain NH2 group of Asn (on the flexible Lid2) forming the oxyanion hole. Hydrogen bonds are represented by dashed lines.

catalytic triad machinery and proposes a "catalytic site tuning" mechanism underlying promiscuous catalysis, expanding the current mechanistic paradigm of enzyme promiscuity.

## Methods

### Gene cloning

The DNA fragments encoding ValDLT (residues 1–418, WP_017819748.1 [https://www.ncbi.nlm.nih.gov/protein/WP_017819748.1]), lec/VC_A0218 (residues 1–418, WP_000746867.1 [https://www.ncbi.nlm.nih.gov/protein/WP_000746867.1]) and VPA0226 (residues 2–418, WP_017448010.1 [https://www.ncbi.nlm.nih.gov/protein/WP_017448010.1]) were amplified via PCR method, using the genomic DNA of *V. alginolyticus* (ATCC 17749), *V. cholerae* O1 biovar EI Tor, *V. parahaemolyticus* (ATCC 17802) as the templates, respectively. These DNA fragments were inserted into the expression vector pETM13 (European Molecular Biology Laboratory) using NcoI and EcoRI restriction sites. The expression construct for VvPlpA (residues 1–417, WP_017420102.1 [https://www.ncbi.nlm.nih.gov/protein/WP_017420102.1]) had been previously prepared at our lab[20]. The mutants were prepared by an overlapping PCR protocol. All the constructs contained a 6xHis tag at their C-termini to facilitate purification. Primers used in this study have been listed in Supplementary Table 1.

### Protein expression and purification

All proteins were produced through the same expression and purification procedure. The sequence-verified constructs were transformed into *E. coli* C43 (DE3). The cells were cultured in Luria−Bertani medium (supplemented with 50 μg/ml kanamycin) at 37 °C until $OD_{600}$ approached ~0.8. The temperature was then shifted to 16 °C, and the cells were induced by 0.2 mM isopropyl β-D-1-thiogalactopyranoside (IPTG) for 16 h. Then, the cells were harvested by centrifugation, resuspended in lysis buffer containing 50 mM Tris-HCl, pH 8.0, and 150 mM NaCl, and lysed by sonication. The insoluble pellets were removed by centrifugation at $15,000 \times g$ for 1 h. The supernatant was subsequently loaded onto a Ni-NTA column (Ni-NTA, BBI) pre-equilibrated in lysis buffer. Then the column was washed with lysis buffer supplemented with 20 mM imidazole. The protein was eluted with lysis buffer supplemented with 250 mM imidazole. Thereafter, the protein sample was subjected to size-exclusion chromatography on the column Superdex-75 16/600 (GE Healthcare) equilibrated in a buffer consisting of 10 mM HEPES pH 7.5, 150 mM NaCl, and 1 mM DTT. The purified proteins were concentrated to 10−20 mg/ml as determined by absorbance at 280 nm and stored in aliquots at −80 °C. The protein folding states were evaluated by thermal shift assay.

### Crystallization, diffraction, and structure determination

Crystallization was performed using the hanging drop vapor diffusion method at 20 °C. ValDLT crystals were grown by mixing 1.0 μl of protein solution (10 mg/ml in the buffer 10 mM HEPES pH 7.5, 150 mM NaCl, 1 mM DTT, 200 mM NDSB-201) and 1.0 μl of reservoir solution (20.5% PEG 4000, 20% PEG 400, 0.1 M Tris pH 8.5 and 0.1 M $MgCl_2$). Crystals of the ValDLT/LAA complex were grown from the drops containing 1.0 μl of protein solution (10 mg/ml in the buffer 10 mM HEPES pH 7.5, 150 mM NaCl, 1 mM DTT, 200 mM NDSB-201 and 3 mM sodium laurate, incubated at 4 °C overnight) and 1.0 μl of reservoir solution (17.5% PEG 3350, 0.2 M magnesium formate). Crystals of the ValDLT/OLA complex were grown from drops containing 1.0 μl of protein solution (5 mg/ml in the buffer 10 mM HEPES pH 7.5, 150 mM NaCl, 1 mM DTT, 300 mM NDSB-201 and 1 mM oleoyl-CoA, incubated at 25 °C for 2 h) and 1.0 μl of reservoir solution (22% PEG 3350, 0.15 M magnesium formate). Crystals of the ValDLT/ARA complex were grown from drops containing 1.0 μl of protein solution (6 mg/ml in the buffer 10 mM HEPES pH 7.5, 150 mM NaCl, 1 mM DTT, 300 mM NDSB-201 and 1.25 mM ARA, incubated at 25 °C for 2 h) and 1.0 μl of reservoir solution

(25% PEG 20000, 0.15 M magnesium formate). Crystals of the ValDLT/DHA complex were grown from drops containing 1.0 μl of protein solution (6 mg/ml in the buffer 10 mM HEPES pH 7.5, 150 mM NaCl, 1 mM DTT, 300 mM NDSB-201 and 1.25 mM DHA, incubated at 25 °C for 2 h) and 1.0 μl of reservoir solution (25% PEG 20000, 0.15 M magnesium formate). The crystallization solution with an addition of 15% PEG 400 served as the cryoprotectant buffer for the ValDLT/LAA and ValDLT/OLA crystals. The cryoprotectant buffer for ValDLT/ARA and ValDLT/DHA is 20% PEG 20000, 0.15 M magnesium formate, 15% PEG 400, supplemented with 2.5 mM ARA or DHA, respectively. Suitable crystals were soaked briefly in the cryoprotectant buffer and then quickly frozen with liquid nitrogen. The *apo* form of ValDLT crystals were frozen with liquid nitrogen directly.

The diffraction data were collected at 100 K on the beamlines BL18U1 and BL19U1 of Shanghai Synchrotron Radiation Facility, using the Blu-Ice software (SSRF)[44]. The data were processed by autoPROC calling the programs XDS and Aimless[45–47]. The structure of *apo* ValDLT was solved by molecular replacement using Phaser[48], with the structure of VvPlpA (6JKZ) as search model. Refinement of the atomic coordinates, B-factors, and TLS parameters using autoBUSTER[49] and model building using Coot[50] were carried out alternately. The ligand-bound structures were solved by molecular replacement using the *apo* structure as search model, being further refined and built with a protocol similar to that for the *apo* form. Ligands were modeled at late stages of refinement, and the geometry restraints of ligands were generated by the GRADE server (http://grade.globalphasing.org). The MolProbity server[51] and other programs in the CCP4 package[52] were also used for structure analysis. PyMOL (Schrödinger) was used for preparing structural graphics. All the crystallographic data are summarized in Supplementary Table 2.

### Esterase assay

Esterase activity of ValDLT was measured by using p-nitrophenyl esters (including pNP-ACA, pNP-BUA, pNP-OCA, pNP-LAA, pNP-PAA or pNP-OLA) as substrates, and detected on a Cary 60 UV-Vis spectrophotometer (Agilent Technologies) with 10-mm path length cuvettes. The hydrolysis of p-nitrophenyl esters was conducted at 25 °C and the increase of product p-nitrophenol was recorded by monitoring the absorbance at a wavelength of 405 nm. To measure the specific activity, the assays were carried out in a 350 μl reaction system contained 150 mM NaCl, 0.3% Triton X-100 and 400 μM substrates and 1.183 μM ValDLT or its mutants. For a kinetics measurement, the assays were carried out in a 350 μl reaction system contained 150 mM NaCl, 0.3% Triton X-100 and a series concentration of pNP-LAA (16−400 μM) and 0.5915 μM ValDLT. A standard curve of the product p-nitrophenol was drawn to convert absorbance values to p-nitrophenol concentrations. Kinetics parameters were estimated by nonlinear fitting of the initial velocity versus substrate concentration data to the Michaelis−Menten equation ($v = V_{max}[S]/(K_M + [S])$) using the Origin program.

### Transferase assay

The assay was performed in reference to a previous protocol[11]. Briefly, 50 μl of cholesterol solution (20 mM, dissolved in chloroform) and 50 μl cardiolipin (18:1) (20 mM, dissolved in chloroform) were mixed together and dried to form lipid films. Then the films of lipid mixtures were hydrated in 1 ml of 50 mM Tris-HCl, pH 7.4, 160 mM KCl by continuous vortex for 5 min and taken through five cycles of freezing-thawing in liquid nitrogen and a warm water bath at 37 °C. The lipids were extruded 21 times through the 100 nm polycarbonate membranes using an Avanti Miniextruder. 10 μl of above liposomes containing 200 μM cholesterol:cardiolipin (18:1) was mixed with 5 μg of purified ValDLT, ValDLT^H393A, VPA0226, lec/VC_A0218 or VvPlpA in 10 mM HEPES, 150 mM NaCl, pH 7.5, plus 1.4% fat-free albumin in a total reaction volume of 20 μl and incubated at 37 °C for 2 h. The lipids were

extracted using the Bligh and Dyer's method and separated by thin-layer chromatography on TLC Silica gel 60 plates (Sigma) with a mobile phase system containing petroleum ether: ethyl ether: acetic acid in a ratio of 90:10:1. The separated lipid spots were visualized by exposure to iodine vapor.

### Metal ion-binding affinity for ValDLT

We used a titration assay to estimate the metal ion-binding affinity for ValDLT referring to a previous protocol[53]. In brief, the enzymatic activity was measured in a 350 μl reaction system contained 150 mM NaCl, 0.3% Triton X-100, 400 μM pNP-LAA and 0.5915 μM ValDLT, supplemented with varying concentrations (0–200 mM) of metal ion ($MgCl_2$, $CaCl_2$ or $MnCl_2$). The changes of enzyme initial velocity ($\Delta v$) were calculated and plotted against the concentrations of metal ions. The apparent $K_D$ value was estimated by fitting to the formula $\Delta v = \Delta V_{max} [S]/(K_D(\text{apparent}) + [S])$ with Origin program.

### Thermal stability evaluation for purified proteins

We use thermal shift assay[54] to monitor the thermal stability of purified ValDLT and its mutants, to guarantee that the proteins are folded well. The thermal shift assay was performed on a quantitative PCR instrument. For each reaction, 12.5 μl of protein solution (0.2 mg/ml in 10 mM HEPES, pH 7.5, 150 mM NaCl buffer) was transferred to a transparent PCR plate, followed by addition of 12.5 μl of SYPRO Orange dye (Sigma, 20× in 10 mM HEPES, pH 7.5, 150 mM NaCl). The mixture was incubated at room temperature for 30 min before fluorescence detection using a fluorescence quantitative PCR instrument. The reaction was measured as a function of temperature at a climbing rate of 1 °C/min from 25 to 95 °C. The $T$m value was calculated by using the Boltzmann equation in Origin program.

### Reporting summary

Further information on research design is available in the Nature Portfolio Reporting Summary linked to this article.

## Data availability

The coordinates and diffraction data generated in this study have been deposited in the PDB (www.rcsb.org) with accession numbers of 8H09 (*apo* ValDLT), 8H0A (ValDLT/LAA), 8H0B (ValDLT/OLA), 8H0C (ValDLT/ARA), and 8H0D (ValDLT/DHA). The crystal structure of VvPlpA used in this study is available in the PDB under the accession code 6JKZ. All other data are contained within the manuscript and the supplementary information file. Source data are provided with this paper.

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

## Acknowledgements

We thank Xuecui Zhao and Prof. Shengbiao Wan for providing chemical compounds, Prof. Rongguang Zhang and Dr. Deqiang Yao for critical reading of the manuscript, Dr. Ioannis Riziotis for discussing catalytic site flexibility and Dr. Xin Wang and Ms. Mengxue Han for help in writing and figure preparation. We also thank the staff from the BL18U1 and BL19U1 beamlines at Shanghai Synchrotron Radiation Facility for assistance in data collection. This work was supported by "Qingdao Innovation Leadership Program" (18-1-2-12-zhc to Q.M.) and "Aoshan Talents Program" of the Pilot National Laboratory for Marine Science and Technology (2015ASTP to Q.M.).

## Author contributions

C.W., C.L., X.Z., and Q.M. performed the experiments. C.W., C.L., and Q.M. analyzed the data and wrote the paper. Q.P. contributed the analytic method. Q.M. conceived and supervised the study. All authors reviewed the results and approved the final version of the manuscript.

## Competing interests

The authors declare no competing interests.
