## [Peer Review File · Nature Communications]

Catalytic site flexibility facilitates the substrate and catalytic promiscuity of Vibrio dual lipase/transferaseREVIEWER COMMENTS

Reviewer #1 (Remarks to the Author):

The investigators present a very interesting structural/mechanistic study in which they describe the structural basis for enzyme promiscuity, in terms of both substrate and catalysis, in a lipase from a pathogenic bacterium that harbors a classic ser-his-asp catalytic triad. They argue that substrate promiscuity and catalytic promiscuity are enabled by the flexibility of key regions and demonstrate that formation of the hallmark canonical "oxyanion hole" is substrate dependent and makes it possible for the enzyme to act as an esterase with various substrates, or even a transferase. This conclusion is based primarily on a thorough comparison of structures the enzyme with fatty acids of varying length, and in the apo form. The idea that this enzyme is poised to hydrolyze a variety of phospholipids is attractive, given that phospholipid content can vary tremendously. Overall, the manuscript is clearly written. However, it would benefit from editing to remove minor language errors.

In terms of the experimental content, this reviewer has one significant concern that the conclusion about the different binding modes of the fatty acids may be overstated/oversimplified. The investigators argue that fatty acids differ significantly in their binding modes, yet in Fig 4 there seems to be only 2 binding modes, and only lauric acid, the saturated 12C fatty acid adopts a unique position. The motivation for lauric acid was that the p-Nitrophenol conjugate was a better substrate than the esters with shorter hydrocarbon chains. It was also a better substrate than the palmitate analog. No comparable data is presented for the unsaturated fatty acids reported in complex with the enzyme, all of which conform nicely the contours of the acyl binding site with their carboxylates superimposed. Is the activity with the pNp-lauric acid mimicking a biological reaction? If not, then is this "catalytic tuning" just something that can be optimized for biotech reasons rather than a characteristic of this enzyme? Is it required for virulence?

Reviewer #2 (Remarks to the Author):

The authors describe catalytic site flexibility in *Vibrio thermolabile* hemolysin and suggest the active site plasticity allows for enzyme promiscuity. They provide five structures of the enzyme in the apo and fatty acid bound states (LAA, OLA, ARA, and DHA). The structures reveal an intrinsically flexible oxyanion hole linked to a mobile lid at the opening of the substrate binding pocket that undergoes disorder to order conformation change upon binding substrates. The ordered lids differ between the LAA bound structure and the long chain fatty acid structures (OLA, ARA, and DHA), and the hydrophobic tunnel accommodates the various substrate lengths. Overall, the observations of plasticity and relationship to enzyme promiscuity is quite interesting. However, the non-native English writing style is difficult to understand throughout the paper and should be addressed for publication.

VPA0226 has several alternate names. It was first named based on demonstration of a haemolytic factor from *V. parahaemolyticus* activated by the addition of lecithin and was denoted as lecithin-dependent haemolysin (LDH). Subsequent cloning of the activity designated the name thermolabile haemolysin to the gene product. Due to this ambiguity, we request the authors include the ordered locus "VPA0226" in their reference to the type 2 secreted lipase that was shown to esterify cholesterol with host fatty acids as this is the most current name used and does not allow ambiguity for its mechanism.

p. 5 "Such a structural design is quite unusual among the enzymes that employ the same catalytic machinery" references quite an old paper. I believe similar enzymes can have mobile catalytic sites, even in the SGNH/GDSL hydrolase family. For example, the SGNH-type acetyl xylan esterases, although working on a different substrate, are reported to have a flexible active site loop (including the His and Asp) that plays a role in cold adaptation of the enzyme. In bacterial acetylcholinesterase structures, the catalytic Ser adopts productive and nonproductive conformations depending on substrate binding, and the paper includes a list of enzymes with similar catalytic site plasticity (PMID: 32371400). The authors should probably exclude terms like "novel" on p.2 and p.12 and change the wording of the above sentence to include potential

exceptions to mobile catalytic machinery.

p. 6 and Fig 3D. similar substrate binding tunnel adaptation (flipping of F338) has been observed in species specific SGNH enzyme mammalian lipopolysaccharide detoxifier (PMID: 29343645). It is interesting that a similar adaptation occurs within the same enzyme in the case of *Vibrio* thermolabile hemolysin. I also wonder if comparison to the LPS detoxifier might give some clues to the cholesterol binding site or if mapping the conservation to the structure might better highlight any potential cholesterol binding residues?

p. 7 chemically homologous molecules should be chemically analogous molecules (and thereafter when referring to a chemical compound).

p. 8 "using all available structures" is ambiguous. Please elaborate on which structures.

The authors make an intriguing point about the mobile site allowing a tradeoff for activity and substrate preference, specifically in reference to the increased activity seen with the mutant ValTLHG203A on pNP-OCA, which has a cholesterol-like backbone. A few mutants were tested on both hydrolysis and transferase activity, but this mutant was excluded. It would be nice to see the hydrolase/transferase activity preference for this mutation.

Reviewer #3 (Remarks to the Author):

In this manuscript, Wang and coworkers present key insights into the substrate and catalytic promiscuity of thermolabile hemolysins (THLs). Guided by crystal structures of the apo enzyme, as well as THL-substrate/product complexes, the authors provide evidence that the promiscuity of the enzyme is the result of different substrate binding modes and an intrinsically flexible oxyanion hole. Additionally, the study is supported by detailed kinetic characterization of variants, which further shed light on the role of metal-chelating residues and the side chains that make up the oxyanion hole.

While this reviewer is not an expert for THLs, this appears to be a fine study, which sheds light on the molecular aspects of catalytic/substrate promiscuity in an important class of enzymes. The findings that a flexible oxyanion hole can be key to determining whether hydrolase or transferase activity is observed could be a rather general mechanism. Particularly, for researchers that aim to use lipases for synthetic purposes, these findings might be particularly notable.

Overall, I believe that this manuscript is suitable for publication in Nature Communications if the following remarks/questions are addressed:

1. This reviewer is curious why transferase activity assays have only been performed with cardiolipin and not with oleic acid and similar p-NO₂-phenol esters. In all cases the same acyl-enzyme intermediate is formed and thus the reactivity should be analogous, unless cardiolipin itself has a role in binding of the cholesterol. Testing whether transferase activity can be observed for p-NO₂-phenol esters should therefore be a fast and straightforward means to assess the role of cardiolipin for transferase activity.

2. Page 6, Line 153: The authors highlight a disorder-to-order conformation for monomer A in presence of LAA. However, it would be interesting to see if there are also significant changes with respect to monomer B of the apo enzyme, which had a structured oxyanion hole. Are these conformations comparable? Similarly, does monomer B differ in the two structures, as LAA is not bound in this monomer?

3. Figure S2: The kinetic measurements as well as the binding affinity measurements do not feature error bars for the individual points. The figure caption mentions that the data points are the averages of three independent measurements, but standard deviations for data points are missing.

4. Page 7, Line 187: When discussing the transferase activity, a reaction scheme would be helpful that shows the reaction. The average reader might not be familiar with structures such as cardiolipin.

5. Page 8, Line 254: The authors discuss plausible residues that could make up the cholesterol

binding pocket. Have the authors attempted to dock cholesterol into their structure featuring oleic acid? This might further pinpoint the exact location of the second substrate.

REVIEWER COMMENTS

Reviewer #1 (Remarks to the Author):

The investigators present a very interesting structural/mechanistic study in which they describe the structural basis for enzyme promiscuity, in terms of both substrate and catalysis, in a lipase from a pathogenic bacterium that harbors a classic ser-his-asp catalytic triad. They argue that substrate promiscuity and catalytic promiscuity are enabled by the flexibility of key regions and demonstrate that formation of the hallmark canonical "oxyanion hole" is substrate dependent and makes it possible for the enzyme to act as an esterase with various substrates, or even a transferase. This conclusion is based primarily on a thorough comparison of structures the enzyme with fatty acids of varying length, and in the apo form. The idea that this enzyme is poised to hydrolyze a variety of phospholipids is attractive, given that phospholipid content can vary tremendously. Overall, the manuscript is clearly written. However, it would benefit from editing to remove minor language errors.

In terms of the experimental content, this reviewer has one significant concern that the conclusion about the different binding modes of the fatty acids may be overstated/oversimplified. The investigators argue that fatty acids differ significantly in their binding modes, yet in Fig 4 there seems to be only 2 binding modes, and only lauric acid, the saturated 12C fatty acid adopts a unique position. The motivation for lauric acid was that the p-Nitrophenol conjugate was a better substrate than the esters with shorter hydrocarbon chains. It was also a better substrate than the palmitate analog. No comparable data is presented for the unsaturated fatty acids reported in complex with the enzyme, all of which conform nicely the contours of the acyl binding site with their carboxylates superimposed. Is the activity with the pNp-lauric acid mimicking a biological reaction? If not, then is this "catalytic tuning" just something that can be optimized for biotech reasons rather than a characteristic of this enzyme? Is it required for virulence?

Reviewer #2 (Remarks to the Author):

The authors describe catalytic site flexibility in *Vibrio thermolabile* hemolysin and suggest the active site plasticity allows for enzyme promiscuity. They provide five structures of the enzyme in the apo and fatty acid bound states (LAA, OLA, ARA, and DHA). The structures reveal an intrinsically flexible oxyanion hole linked to a mobile lid at the opening of the substrate binding pocket that undergoes disorder to order conformation change upon binding substrates. The ordered lids differ between the LAA bound structure and the long chain fatty acid structures (OLA, ARA, and DHA), and the hydrophobic tunnel accommodates the various substrate lengths. Overall, the observations of plasticity and relationship to enzyme promiscuity is quite interesting. However, the non-native English writing style is difficult to understand throughout the paper and should be addressed for publication.

VPA0226 has several alternate names. It was first named based on demonstration of a haemolytic

factor from *V. parahaemolyticus* activated by the addition of lecithin and was denoted as lecithin-dependent haemolysin (LDH). Subsequent cloning of the activity designated the name thermolabile haemolysin to the gene product. Due to this ambiguity, we request the authors include the ordered locus “VPA0226” in their reference to the type 2 secreted lipase that was shown to esterify cholesterol with host fatty acids as this is the most current name used and does not allow ambiguity for its mechanism.

p. 5 “Such a structural design is quite unusual among the enzymes that employ the same catalytic machinery” references quite an old paper. I believe similar enzymes can have mobile catalytic sites, even in the SGNH/GDSL hydrolase family. For example, the SGNH-type acetyl xylan esterases, although working on a different substrate, are reported to have a flexible active site loop (including the His and Asp) that plays a role in cold adaptation of the enzyme. In bacterial acetylcholinesterase structures, the catalytic Ser adopts productive and nonproductive conformations depending on substrate binding, and the paper includes a list of enzymes with similar catalytic site plasticity (PMID: 32371400). The authors should probably exclude terms like “novel” on p.2 and p.12 and change the wording of the above sentence to include potential exceptions to mobile catalytic machinery.

p. 6 and Fig 3D. similar substrate binding tunnel adaptation (flipping of F338) has been observed in species specific SGNH enzyme mammalian lipopolysaccharide detoxifier (PMID: 29343645). It is interesting that a similar adaptation occurs within the same enzyme in the case of *Vibrio thermolabile* hemolysin. I also wonder if comparison to the LPS detoxifier might give some clues to the cholesterol binding site or if mapping the conservation to the structure might better highlight any potential cholesterol binding residues?

p. 7 chemically homologous molecules should be chemically analogous molecules (and thereafter when referring to a chemical compound).

p. 8 “using all available structures” is ambiguous. Please elaborate on which structures.

The authors make an intriguing point about the mobile site allowing a tradeoff for activity and substrate preference, specifically in reference to the increased activity seen with the mutant ValTLHG203A on pNP-OCA, which has a cholesterol-like backbone. A few mutants were tested on both hydrolysis and transferase activity, but this mutant was excluded. It would be nice to see the hydrolase/transferase activity preference for this mutation.

Reviewer #3 (Remarks to the Author):

In this manuscript, Wang and coworkers present key insights into the substrate and catalytic promiscuity of thermolabile hemolysins (THLs). Guided by crystal structures of the apo enzyme, as well as THL-substrate/product complexes, the authors provide evidence that the promiscuity of the enzyme is the result of different substrate binding modes and an intrinsically flexible oxyanion hole. Additionally, the study is supported by detailed kinetic characterization of variants, which further shed light on the role of metal-chelating residues and the side chains that make up the oxyanion hole.

While this reviewer is not an expert for THLs, this appears to be a fine study, which sheds light on

the molecular aspects of catalytic/substrate promiscuity in an important class of enzymes. The findings that a flexible oxyanion hole can be key to determining whether hydrolase or transferase activity is observed could be a rather general mechanism. Particularly, for researchers that aim to use lipases for synthetic purposes, these findings might be particularly notable.

Overall, I believe that this manuscript is suitable for publication in Nature Communications if the following remarks/questions are addressed:

1. This reviewer is curious why transferase activity assays have only been performed with cardiolipin and not with oleic acid and similar p-NO₂-phenol esters. In all cases the same acyl-enzyme intermediate is formed and thus the reactivity should be analogous, unless cardiolipin itself has a role in binding of the cholesterol. Testing whether transferase activity can be observed for p-NO₂-phenol esters should therefore be a fast and straightforward means to assess the role of cardiolipin for transferase activity.

2. Page 6, Line 153: The authors highlight a disorder-to-order conformation for monomer A in presence of LAA. However, it would be interesting to see if there are also significant changes with respect to monomer B of the apo enzyme, which had a structured oxyanion hole. Are these conformations comparable? Similarly, does monomer B differ in the two structures, as LAA is not bound in this monomer?

3. Figure S2: The kinetic measurements as well as the binding affinity measurements do not feature error bars for the individual points. The figure caption mentions that the data points are the averages of three independent measurements, but standard deviations for data points are missing.

4. Page 7, Line 187: When discussing the transferase activity, a reaction scheme would be helpful that shows the reaction. The average reader might not be familiar with structures such as cardiolipin.

5. Page 8, Line 254: The authors discuss plausible residues that could make up the cholesterol binding pocket. Have the authors attempted to dock cholesterol into their structure featuring oleic acid? This might further pinpoint the exact location of the second substrate.

Response to Reviewers' Comments

We sincerely thank all the editors and reviewers for the time and effort in evaluating our manuscript. The constructive and detailed comments are very helpful in improving the manuscript quality. Below is our point-by-point response, with the reviewers' original comments in regular black text, our response in blue and the final words appearing in the revised manuscript in red. The page and line numbers are referred to those in the track-changed revision file.

Reviewer #1 (Remarks to the Author):

The investigators present a very interesting structural/mechanistic study in which they describe the structural basis for enzyme promiscuity, in terms of both substrate and catalysis, in a lipase from a pathogenic bacterium that harbors a classic ser-his-asp catalytic triad. They argue that substrate promiscuity and catalytic promiscuity are enabled by the flexibility of key regions and demonstrate that formation of the hallmark canonical "oxyanion hole" is substrate dependent and makes it possible for the enzyme to act as an esterase with various substrates, or even a transferase. This conclusion is based primarily on a thorough comparison of structures the enzyme with fatty acids of varying length, and in the apo form. The idea that this enzyme is poised to hydrolyze a variety of phospholipids is attractive, given that phospholipid content can vary tremendously. Overall, the manuscript is clearly written. However, it would benefit from editing to remove minor language errors.

We sincerely thank the reviewer for the positive evaluation of our work. To remove the language errors, the revision has been edited by an expert language company.

In terms of the experimental content, this reviewer has one significant concern that the conclusion about the different binding modes of the fatty acids may be overstated/oversimplified. The investigators argue that fatty acids differ significantly in their binding modes, yet in Fig 4 there seems to be only 2 binding modes, and only lauric acid, the saturated 12C fatty acid adopts a unique position. The motivation for lauric acid was that the p-Nitrophenol conjugate was a better substrate than the esters with shorter hydrocarbon chains. It was also a better substrate than the palmitate analog. No comparable data is presented for the unsaturated fatty acids reported in complex with the enzyme, all of which conform nicely the contours of the acyl binding site with their carboxylates superimposed. Is the activity with the pNp-lauric acid mimicking a biological reaction? If not, then is this "catalytic tuning" just something that can be optimized for biotech reasons rather than a characteristic of this enzyme? Is it required for virulence?

We thank the reviewer for bringing about this important and interesting issue. Here we would provide extra arguments to further clarify our opinion on fatty acid/acyl binding mode diversity. To better illustrate the acyl chain length and saturation, we will use carbon numbering annotation for fatty acids in the response: C8 for octanoic acid, C12 for lauric acid, C16 for palmitic acid, C18:1 for oleic acid, C20:4 for arachidonic acid, and C22:6 for docosahexaenoic acid.

First of all, we agree with the reviewer that the binding modes of the four fatty acids can be roughly divided into two categories, although their zigzag and interaction details with the pocket differ

from each other (Fig. R1a and b). To be more accurate and specific, we rephrase the description of fatty acid binding modes. The text “The acyl chains of different fatty acids accommodated in this pocket display various gestures and differently interact with the surrounding hydrophobic residues, while their carboxyl groups are positioned at the opening of the acyl-binding pocket with varying conformations as well (Fig. 4B).” has been now replaced by:

“The fatty acid ligands are accommodated in this acyl-binding pocket in different poses, displaying different interactions with the surrounding residues (Fig. 4a and Supplementary Fig. 5). The hydrocarbon tails of OLA, ARA and DHA, which are unsaturated long-chain fatty acids, all adopt a bent conformation to occupy a large portion of the pocket. Their tails follow a similar tracking path with certain differences in zigzag detail, and their carboxyl groups are in the same position with similar orientation. In contrast, LAA, a saturated medium-chain fatty acid, has its hydrocarbon tail in an extended conformation, in a space different from the previously discussed fatty acids; the carboxyl group of LAA is located at a position obviously different from that of the former three fatty acids.” (Page 10, Line 315)

Fig. R1. Structural analysis of fatty acid (acyl chain) binding modes in the acyl-binding pocket. (a) The acyl-binding pocket subzones (I, II, III, IV) and the binding modes of the four fatty acids, C12, C18:1, C20:4, and C22:6. (b) Details of the interactions between the fatty acids and pocket residues. The mobile Lid1, Lid2 and inner Phe lid are indicated. Subzones are pointed as well. (c) Chemical structure of the fluorescent substrate Red/Green BODIPY® PC-A2.

Regarding other fatty acids (acyl chains), we propose that they may have a binding mode similar to the aforementioned four fatty acids or even a new binding mode beyond what is observed in this study. There are a couple of structural and experimental clues to support this proposal. Firstly, we take C16 as an example. Despite extensive efforts, we have not obtained crystals of the ValTLH/C16 complex yet, including those conditions for the four fatty acids. Although crystallization can be affected by many factors, the ill crystallization behavior of the ValTLH/C16 complex could be a sign of different binding of C16 to the protein. As shown in Fig. R1a and b, the acyl-binding pocket of TLH can be roughly subdivided into four subzones (I, II, III, and IV) and there are a couple of possible binding modes for C16. C16 could resemble C12 to sit in subzones I/II, but its chain has to take a rather curly conformation. C16 could also pass the Phe lid to locate in subzones I/IV, as the unsaturated long-chain fatty acids C18:1, C20:4, and C22:6. Because of lacking any double bond, C16 would have a very “soft” chain and thus could interact differently with the pocket than the unsaturated long-chain fatty acids. Or, the “soft” tail of C16 could even occupy the place of subzones I and III, different from the four fatty acids reported in this study. Secondly, we take Red/Green BODIPY PC-A2 as another example (Fig. R1c), which is an artificial fluorescent substrate used for measuring the phospholipase A2 activity of TLHs. It has a bulky acyl chain at the sn-2 position, which needs to be buried into the pocket for catalysis. We think the pocket may undergo certain shape reformation to accommodate such a bulky acyl chain, which is structurally possible because a part of the pocket wall is contributed by the mobile Lid2 (Fig. R1a). Thirdly, although unverified, the acyl binding mode of a lipid substrate might also be affected by the diversity in its other acyl chain and head group (e.g. phosphatidylcholine versus cardiolipin), considering the prominent structural flexibility of the enzyme. Last, we would recapitulate the micro-perturbation experiment (Fig. 4f in revision), in which the substrates (pNP-C8, -C12, and -C18:1) responded differently to oxyanion hole region perturbation. This suggests that differential configurations of the oxyanion region are required for optimally stabilizing their oxyanion, reflecting different positioning of their reactive –COO– groups. Otherwise, equally “better” or “worse” effects are expected for these analogous substrates. Taken together, we propose that the fatty acid/acyl binding modes may display certain diversity. Accordingly, the –COO– ester groups and the developed oxyanions could be positioned in a substrate-dependent manner, rationalizing why TLHs evolve the catalytic site tuning mechanism. In the future, we would seek to solve more complex structures to better address the binding mode issue.

(Is the activity with the pNP-lauric acid mimicking a biological reaction? If not, then is this “catalytic tuning” just something that can be optimized for biotech reasons rather than a characteristic of this enzyme?)

In this study, we characterized the enzyme using a couple of common substrates. So far, the *in vivo* substrate spectra of TLHs are not completely clear yet, but we think the activity on pNP-C12 may have certain biological relevance. We performed experiment to show that ValTLH can hydrolyze the lipid 1,2-dilauroyl-sn-glycero-3-phosphocholine (DLPC) (Fig. R2a). Although this lipid is not abundant in human cell membranes, it still can be a natural substrate for TLH, considering that vibrio bacteria are opportunistic pathogens residing in a wide range of hosts and living together with the complex marine biota. Interestingly, our experiment showed that TLH can also hydrolyze C12-CoA (Fig. R2b), a rather common intermediate in the fatty acid metabolism pathway of the cell

(For example, fatty acyl-CoA synthase can catalyze the esterification of fatty acids with CoA at the outer membrane of mitochondria). Although this acyl-CoA hydrolysis activity is not investigated *in vivo* yet, implicitly, TLH could also interfere with the fatty acid metabolism/lipid biosynthesis of the host cell.

Fig. R2. ValTLH hydrolyzed DLPC and C12-CoA. (A) The fatty acid released from DLPC hydrolysis was tracked by monitoring pH changes. (B) The CoA released from C12-CoA hydrolysis was detected by a fluorescence dye.

As for the physiological relevance of catalytic site tuning, we think the bacteria evolve this mechanism to benefit their own fitness. *Vibrio* are opportunistic pathogens living in both hosts and aqua environments. TLHs can encounter a large array of lipidic substrates with complex and diverse structures, which exhibit certain binding diversity and thus the oxyanion positioning variations. Catalytic site tuning would endow the enzymes with the ability to handle not only the “big” oxyanion positioning variations due to the substrate binding differences, but also the “small” oxyanion positioning fluctuation along the catalysis process. Therefore, we think catalytic site tuning is a characteristic of the TLH enzymes. As a bonus, this interesting characteristic of TLHs would facilitate their use in biotech applications.

(Is it required for virulence?)

With respect to the requirement of catalytic site tuning for virulence, we don’t have an exact answer yet for lacking experimental evidence. We speculate two aspects that catalytic site tuning may contribute to virulence. The key value of catalytic site tuning is to endow TLH with versatility (promiscuity) in lipid processing. With such a mechanism, the pathogen can use a single versatile enzyme instead of multiple specific enzymes in processing the diverse host lipids, being economical yet efficient. Generally, *vibrio* pathogens can infect a wide range of host organisms. Using the catalytic site tuning mechanism, TLH can readily self-adapt to the specific lipid pool of a given host, which saves a complex sensing-responding circuit in pathogens, being simple and robust. These advantages of catalytic site tuning partially explain why the pathogen evolves such an interesting enzyme mechanism.

Reviewer #2 (Remarks to the Author):

The authors describe catalytic site flexibility in *Vibrio thermolabile* hemolysin and suggest the active site plasticity allows for enzyme promiscuity. They provide five structures of the enzyme in the apo and fatty acid bound states (LAA, OLA, ARA, and DHA). The structures reveal an intrinsically flexible oxyanion hole linked to a mobile lid at the opening of the substrate binding pocket that undergoes disorder to order conformation change upon binding substrates. The ordered lids differ between the LAA bound structure and the long chain fatty acid structures (OLA, ARA, and DHA), and the hydrophobic tunnel accommodates the various substrate lengths. Overall, the observations of plasticity and relationship to enzyme promiscuity is quite interesting. However, the non-native English writing style is difficult to understand throughout the paper and should be addressed for publication.

We sincerely thank the reviewer for the positive evaluation of our work. The revision has been edited by an expert language company to address the writing style issue.

VPA0226 has several alternate names. It was first named based on demonstration of a haemolytic factor from *V. parahaemolyticus* activated by the addition of lecithin and was denoted as lecithin-dependent haemolysin (LDH). Subsequent cloning of the activity designated the name thermolabile haemolysin to the gene product. Due to this ambiguity, we request the authors include the ordered locus "VPA0226" in their reference to the type 2 secreted lipase that was shown to esterify cholesterol with host fatty acids as this is the most current name used and does not allow ambiguity for its mechanism.

Thanks for clarifying the name issue. As suggested, the name "VPA0226" has been used instead of "VpTLH" throughout the manuscript.

p. 5 "Such a structural design is quite unusual among the enzymes that employ the same catalytic machinery" references quite an old paper. I believe similar enzymes can have mobile catalytic sites, even in the SGNH/GDSL hydrolase family. For example, the SGNH-type acetyl xylan esterases, although working on a different substrate, are reported to have a flexible active site loop (including the His and Asp) that plays a role in cold adaptation of the enzyme. In bacterial acetylcholinesterase structures, the catalytic Ser adopts productive and nonproductive conformations depending on substrate binding, and the paper includes a list of enzymes with similar catalytic site plasticity (PMID: 32371400). The authors should probably exclude terms like "novel" on p.2 and p.12 and change the wording of the above sentence to include potential exceptions to mobile catalytic machinery.

Thanks for pointing out catalytic site mobility in other enzymes. During the literature search, we also found that some enzymes display certain conformational variations in their catalytic sites, which has been briefly mentioned in our discussion part. Nevertheless, to our knowledge, the flexibility of the oxyanion hole residues at a degree as significant as in TLH is rarely reported. As suggested, the unnecessary word "novel" on P2 and P12 has been removed. To further exemplify catalytic site mobility, the sentence "Such a structural design is quite unusual among the enzymes that employ the same catalytic machinery" has been replaced by:

"Interestingly, it has been proposed that SGNH family proteins can exhibit active site flexibility¹⁹, with some reports including flexibility of the catalytic residues. For example, the acetyl-xylan

esterase from an Arctic marine bacterium has the catalytic His and Asp in a long flexible active site loop, which is expected to play a role in cold adaptation of this enzyme²². In the bacterial acetylcholinesterase structure, the catalytic Ser can adopt productive and nonproductive conformations depending on substrate binding²³. Nevertheless, the large-scale flexibility of oxyanion hole residues, as seen in ValTLH, has rarely been reported²⁴." (Page 5, Line 134)

p. 6 and Fig 3D. similar substrate binding tunnel adaptation (flipping of F338) has been observed in species specific SGNH enzyme mammalian lipopolysaccharide detoxifier (PMID: 29343645). It is interesting that a similar adaptation occurs within the same enzyme in the case of *Vibrio thermolabile* hemolysin. I also wonder if comparison to the LPS detoxifier might give some clues to the cholesterol binding site or if mapping the conservation to the structure might better highlight any potential cholesterol binding residues?

We thank the reviewer for pointing out this very interesting similarity shared by different SGNH enzymes. To highlight this similarity, we have added one sentence:

"Interestingly, similar tunnel adaptation of the Phe residue has also been observed in the SGNH family enzyme mammalian lipopolysaccharide detoxifier²⁵." (Page 8, Line 248)

As suggested, we have compared ValTLH and mammalian LPS detoxifier (named acyloxyacyl hydrolase, AOAH). The two enzymes share less than 10% sequence identity, indicating a distant relationship (Fig. R3a). Structurally, only the SGNH/GDSL parts can be roughly aligned, where the catalytic site residues, and interestingly, the tunnel Phe residues can be well superposed (Fig. R3b, c), just as the reviewer has proposed. In contrast to the certain similarity in the acyl-binding pocket, the putative cholesterol binding region of ValTLH deviates significantly from the counterpart region of the mammalian LPS detoxifier (Fig. R3c), which hinders extracting useful information in the context of cholesterol binding.

Fig. R3. Comparison of ValTLH and mouse AOA. (a) Sequence alignment. The catalytic site residues are marked by green triangles. (b) Overall structural superposition of the structures of ValTLH/OLA and mouse AOA/DMPC (PDB ID: 5W7E). (c) Comparison of the active site regions.

p. 7 chemically homologous molecules should be chemically analogous molecules (and thereafter when referring to a chemical compound).

Thanks for pointing out this point. As suggested, the word “analogous” has been used for similar chemical molecules.

p. 8 “using all available structures” is ambiguous. Please elaborate on which structures.

As suggested, we have clarified the structures used for comparison:

“...using all available ValTLH structures, including the *apo* form and its complexes with LAA, OLA, ARA and DHA.”(Page 10, Line 304)

The authors make an intriguing point about the mobile site allowing a tradeoff for activity and substrate preference, specifically in reference to the increased activity seen with the mutant ValTLHG203A on pNP-OCA, which has a cholesterol-like backbone. A few mutants were tested on both hydrolysis and transferase activity, but this mutant was excluded. It would be nice to see the

hydrolase/transferase activity preference for this mutation.

Thanks for pointing out the chemical similarity between cholesterol and pNP-OCA. As suggested, we have performed an assay to test the hydrolase/transferase activity of ValTLH^{G203A}, as shown in Fig. R4. In general, there is no significant difference between ValTLH^{G203A} and the WT in this assay condition, although ValTLH^{G203A} seems to have slightly weaker hydrolase/transferase activity, with a preference plausibly more biased to hydrolase activity.

Fig. R4. Hydrolase/transferase activity of ValTLH and ValTLH^{G203A} mutant.

Reviewer #3 (Remarks to the Author):

In this manuscript, Wang and coworkers present key insights into the substrate and catalytic promiscuity of thermolabile hemolysins (THLs). Guided by crystal structures of the apo enzyme, as well as THL-substrate/product complexes, the authors provide evidence that the promiscuity of the enzyme is the result of different substrate binding modes and an intrinsically flexible oxyanion hole. Additionally, the study is supported by detailed kinetic characterization of variants, which further shed light on the role of metal-chelating residues and the side chains that make up the oxyanion hole.

While this reviewer is not an expert for THLs, this appears to be a fine study, which sheds light on the molecular aspects of catalytic/substrate promiscuity in an important class of enzymes. The findings that a flexible oxyanion hole can be key to determining whether hydrolase or transferase activity is observed could be a rather general mechanism. Particularly, for researchers that aim to use lipases for synthetic purposes, these findings might be particularly notable.

Overall, I believe that this manuscript is suitable for publication in Nature Communications if the following remarks/questions are addressed:

We sincerely thank the reviewer for the positive evaluation of our work.

1. This reviewer is curious why transferase activity assays have only been performed with cadiolipin and not with oleic acid and similar p-NO₂-phenol esters. In all cases the same acyl-enzyme intermediate is formed and thus the reactivity should be analogous, unless cadiolipin itself has a role in binding of the cholesterol. Testing whether transferase activity can be observed for p-NO₂-phenol esters should therefore be a fast and straightforward means to assess the role of cadiolipin for transferase activity.

Thanks for proposing a simple transferase assay. As suggested, we performed an assay in a cholesterol-containing micelle system, using pNP-OLA as the substrate. As shown in Fig. R5, only hydrolysis of pNP-OLA occurred in this assay system. Likely, transferring acyl to cholesterol tends to take place in a liposome system instead of a micelle system. Or even more interestingly, the head group of the lipidic substrate could have an effect on the reaction-type preference, which is worth investigating in the future.

Fig. R5. Evaluation of the transferase/lipase activity of ValTLH on pNP-OLA. ValTLH catalyzed the hydrolysis of pNP-OLA in the presence of cholesterol, while the transferase activity was not observed (lane 5).

2. Page 6, Line 153: The authors highlight a disorder-to-order conformation for monomer A in presence of LAA. However, it would be interesting to see if there are also significant changes with respect to monomer B of the apo enzyme, which had a structured oxyanion hole. Are these conformations comparable? Similarly, does monomer B differ in the two structures, as LAA is not bound in this monomer?

Thanks for the comment. Indeed, the monomers B in the *apo* and all complex structures share a similarity in the active site region, all of which contain a structured oxyanion hole. To clarify this, we have modified Supplementary Fig. 3 in the revision to include the superposition of all monomers B (Fig. R6). The text has also been rephrased:

“The comparison of monomers A and B with those in the *apo* form is shown in Supplementary Fig. 3. In the active site region, ValTLH/LAA monomer A shows clear structural differences in reference to *apo* monomer A. In contrast, the active site region of ValTLH/LAA monomer B is structurally similar to that of *apo* monomer B, with loop^{β8-α5} in an open conformation and the catalytic site residues well superposed. The LAA molecule was only identified in the substrate-binding pocket (hereafter designated acyl-binding pocket) of monomer A (Fig. 2b). Because no LAA was identified in monomer B, hereafter, our structural analysis was based on monomer A only, unless otherwise specified.” (Page 6, Line 152)

Fig. R6 (Supplementary Fig. 3 in revision). Superposition of the overall structures of monomer A (left) and monomer B (right) of apo ValTLH, ValTLH/LAA, ValTLH/OLA, ValTLH/ARA and ValTLH/DHA. Regions with significant conformational differences are indicated. Catalytic residues are shown in stick representation. Mg^{2+} binding sites in monomer B are indicated, with the binding residues shown in stick representation.

3. Figure S2: The kinetic measurements as well as the binding affinity measurements do not feature error bars for the individual points. The figure caption mentions that the data points are the averages of three independent measurements, but standard deviations for data points are missing. Thanks for pointing out this issue. We have updated Supplementary Fig. 2c and d to include error bars (Fig. R7).

Fig. R7 (Supplementary Fig. 2c and d in revision). Kinetic data of ValTLH.

4. Page 7, Line 187: When discussing the transferase activity, a reaction scheme would be helpful that shows the reaction. The average reader might not be familiar with structures such as cardiolipin.

As suggested, a reaction scheme of the transferase activity has been added, with the chemical structure of cardiolipin clearly shown (Supplementary Fig. 4 in the revision).

Fig. R8 (Supplementary Fig. 4 in revision). Schematic description of the TLH-catalyzed acyl transfer from cardiolipin (C18:1) to cholesterol. For simplicity, only one acyl chain at the sn-2 position of cardiolipin (C18:1) is shown for transfer, but other acyl chains could also be transferred.

5. Page 8, Line 254: The authors discuss plausible residues that could make up the cholesterol binding pocket. Have the authors attempted to dock cholesterol into their structure featuring oleic acid? This might further pinpoint the exact location of the second substrate.

Thanks for the advice. We have used three different programs to dock cholesterol to the OLA bound ValTLH structure. As shown in Fig. R9, all programs docked the cholesterol molecule at the positions far away from the active site, which are unlikely to be the true cholesterol binding site. This failure is not surprising, because the fatty-acid bound ValTLH requires additional conformational changes to allocate the cholesterol molecule. A sophisticated docking algorithm considering protein flexibility might give better hints, but currently we do not have such an expertise. To reveal exact cholesterol binding information, we would seek to solve a cholesterol-complexed structure in the future.

Fig. R9. Docking cholesterol to the VaTLH/OLA complex structure. Docking was performed using the programs: (a) DINC (Devaurs et al, 2019), (b) GRAMM (Katzchalski-Katzir et al, 1992), and (c) DOCKTHOR (Guedes et al, 2021).

REVIEWER COMMENTS

Reviewer #1 (Remarks to the Author):

The revised manuscript has addressed my previous concerns.

Reviewer #2 (Remarks to the Author):

The current manuscript is unacceptable.

The authors state below that they will replace VpTLH with VPA0226 and they have not done so. Nor have they cited the VPA0226 paper for its known activities. See their comment below.

This needs to be rectified and then we will rereview their manuscript for other corrections that were stated to be made.

VPA0226 has several alternate names. It was first named based on demonstration of a haemolytic factor from *V. parahaemolyticus* activated by the addition of lecithin and was denoted as lecithindependent haemolysin (LDH). Subsequent cloning of the activity designated the name thermolabile haemolysin to the gene product. Due to this ambiguity, we request the authors include the ordered locus "VPA0226" in their reference to the type 2 secreted lipase that was shown to esterify cholesterol with host fatty acids as this is the most current name used and does not allow ambiguity for its mechanism.

Thanks for clarifying the name issue. As suggested, the name "VPA0226" has been used instead of "VpTLH" throughout the manuscript.

Reviewer #3 (Remarks to the Author):

The authors have addressed my concerns and as initially suggested, the manuscript is suitable for publication in Nature Communications.

A last remark for the authors. In Scheme 5b, side chains and main chains often lack protons, particularly for nitrogen residues. Please modify the figure in order to display the correct protonation state for each residue.

REVIEWER COMMENTS

Reviewer #1 (Remarks to the Author):

The revised manuscript has addressed my previous concerns.

Reviewer #2 (Remarks to the Author):

The current manuscript is unacceptable.

The authors state below that they will replace VpTLH with VPA0226 and they have not done so. Nor have they cited the VPA0226 paper for its known activities. See their comment below.

This needs to be rectified and then we will rereview their manuscript for other corrections that were stated to be made.

VPA0226 has several alternate names. It was first named based on demonstration of a haemolytic factor from *V. parahaemolyticus* activated by the addition of lecithin and was denoted as lecithindependent haemolysin (LDH). Subsequent cloning of the activity designated the name thermolabile haemolysin to the gene product. Due to this ambiguity, we request the authors include the ordered locus “VPA0226” in their reference to the type 2 secreted lipase that was shown to esterify cholesterol with host fatty acids as this is the most current name used and does not allow ambiguity for its mechanism.

Thanks for clarifying the name issue. As suggested, the name “VPA0226” has been used instead of “VpTLH” throughout the manuscript.

Reviewer #3 (Remarks to the Author):

The authors have addressed my concerns and as initially suggested, the manuscript is suitable for publication in Nature Communications.

A last remark for the authors. In Scheme 5b, side chains and main chains often lack protons, particularly for nitrogen residues. Please modify the figure in order to display the correct protonation state for each residue.

Response to Reviewers' Comments

We sincerely thank all the editors and reviewers again for the time and effort in evaluating our manuscript. The constructive comments have helped us further improve our manuscript. Below is our point-by-point response, with the reviewers' original comments in regular black text, our response in **blue** and the final words appearing in the revised manuscript in **red**. The page and line numbers are referred to those in the track-changed revision file.

Reviewer #1 (Remarks to the Author):

The revised manuscript has addressed my previous concerns.

We thank the reviewer for the positive comment.

Reviewer #2 (Remarks to the Author):

The current manuscript is unacceptable.

The authors state below that they will replace VpTLH with VPA0226 and they have not done so. Nor have they cited the VPA0226 paper for its known activities. See their comment below.

This needs to be rectified and then we will rereview their manuscript for other corrections that were stated to be made.

VPA0226 has several alternate names. It was first named based on demonstration of a haemolytic factor from *V. parahaemolyticus* activated by the addition of lecithin and was denoted as lecithindependent haemolysin (LDH). Subsequent cloning of the activity designated the name thermolabile haemolysin to the gene product. Due to this ambiguity, we request the authors include the ordered locus “VPA0226” in their reference to the type 2 secreted lipase that was shown to esterify cholesterol with host fatty acids as this is the most current name used and does not allow ambiguity for its mechanism.

Thanks for clarifying the name issue. As suggested, the name “VPA0226” has been used instead of “VpTLH” throughout the manuscript.

We thank the reviewer for the effort in reviewing our manuscript. First of all, we apologize for our limited understanding and not properly addressing the naming issue in the last response. We agree with the reviewer that the names lecithin dependent haemolysin (LDH) and thermolabile haemolysin (TLH) have certain limitation in accounting for all the functions of this class of enzymes. To remove any mechanism ambiguity and to reflect the most current knowledge of this class of enzymes, we would avoid using the term “thermolabile hemolysin” in the revision, and instead, we use VPA0226-type lipases as a collective name for this class of enzymes and the species-specific lipases are named as appeared in the literature, such as VPA0226 for *V. parahaemolyticus*, lec/VC_A0218 for *V. cholera*, and VvPlpA for *V. vulnificus*. We also name the VPA0226-type lipase from *V. alginolyticus* as ValLip. In such a way, we hope the naming issue has been better addressed. If required, we are willing to do further corrections.

We have cited the VPA0226 paper (Chimalapati et al., 2020 *eLife*) for its known activities: “Interestingly, a recent study by Chimalapati et al. firstly showed that VPA0226, a type 2 secreted lipase that helps *V. parahaemolyticus* escape from the host cell, not only exhibited the traditional lipase activity but also could transfer the acyl chain from various host lipids to cholesterol¹¹.” (Page 3, Line 63)

Reviewer #3 (Remarks to the Author):

The authors have addressed my concerns and as initially suggested, the manuscript is suitable for publication in Nature Communications.

We thank the reviewer for the positive comment.

A last remark for the authors. In Scheme 5b, side chains and main chains often lack protons, particularly for nitrogen residues. Please modify the figure in order to display the correct protonation state for each residue.

Thanks for this important remark. As requested, we have modified Fig. 5b to include the protonation information for each residue (Fig. R1).

Fig. R1 (Fig. 5b in revision). Schematic representation of the promiscuous catalysis process.

REVIEWERS' COMMENTS

Reviewer #2 (Remarks to the Author):

Thank you for citing the VPA0226 paper, please leave this citation on line 55

However, the enzyme family could use a name that is not "hemolysin" as it does not lyse cells.

The family should be called "Vibrio dual lipase/tranferase".

as VPA0226-type lipase does not include the transferase activity.

REVIEWERS' COMMENTS

Reviewer #2 (Remarks to the Author):

Thank you for citing the VPA0226 paper, please leave this citation on line 55

However, the enzyme family could use a name that is not "hemolysin" as it does not lyse cells.

The family should be called "Vibrio dual lipase/tranferase".

as VPA0226-type lipase does not include the transferase activity.

Response to Reviewers' Comments

We sincerely thank all the editors and reviewers again for the time and effort in evaluating our manuscript. The constructive comments have helped us further improve our manuscript. Below is our point-by-point response, with the reviewers' original comments in regular black text, our response in blue. The page and line numbers are referred to those in the track-changed revision file.

Reviewer #2 (Remarks to the Author):

Thank you for citing the VPA0226 paper, please leave this citation on line 55
However, the enzyme family could use a name that is not "hemolysin" as it does not lyse cells.
The family should be called "Vibrio dual lipase/tranferase".
as VPA0226-type lipase does not include the transferase activity.

We thank the reviewer for the effort in reviewing our manuscript. As requested, the citation of VPA0226 paper has been put on line 55. In the revision, we have substituted "*Vibrio* dual lipase/transferase (VDLT)" for "VPA0226-type lipase" throughout the manuscript.